# Transcript isoform sequencing reveals widespread promoter-proximal transcriptional termination in *Arabidopsis*

Quentin Angelo Thomas [1], Ryan Ard [1], Jinghan Liu [1], Bingnan Li[2], Jingwen Wang [2], Vicent Pelechano [2] & Sebastian Marquardt [1✉]

RNA polymerase II (RNAPII) transcription converts the DNA sequence of a single gene into multiple transcript isoforms that may carry alternative functions. Gene isoforms result from variable transcription start sites (TSSs) at the beginning and polyadenylation sites (PASs) at the end of transcripts. How alternative TSSs relate to variable PASs is poorly understood. Here, we identify both ends of RNA molecules in *Arabidopsis thaliana* by transcription isoform sequencing (TIF-seq) and report four transcript isoforms per expressed gene. While intragenic initiation represents a large source of regulated isoform diversity, we observe that ~14% of expressed genes generate relatively unstable short promoter-proximal RNAs (sppRNAs) from nascent transcript cleavage and polyadenylation shortly after initiation. The location of sppRNAs correlates with the position of promoter-proximal RNAPII stalling, indicating that large pools of promoter-stalled RNAPII may engage in transcriptional termination. We propose that promoter-proximal RNAPII stalling-linked to premature transcriptional termination may represent a checkpoint that governs plant gene expression.

[1] Copenhagen Plant Science Centre, Department of Plant and Environmental Sciences, University of Copenhagen, Frederiksberg, Denmark. [2] SciLifeLab, Department of Microbiology, Tumor and Cell Biology, Karolinska Institutet, Solna, Sweden. ✉email: sebastian.marquardt@plen.ku.dk

Organisms respond to changing environments by regulating the transcriptional activity of RNA polymerase II (RNAPII) to adjust gene expression[1]. However, RNAPII transcription generates distinct mRNA and non-coding RNA isoforms derived from a single gene[2]. As alternative gene isoforms may have opposing functions[3], the identification of transcript isoform diversity can be key to resolve molecular mechanisms underpinning causative genetic variation. The selection of transcription start sites (TSSs) represents a key source of transcript heterogeneity. The majority of TSSs map to gene promoter regions, where DNA sequence elements recruit RNAPII to initiate transcription in nucleosome-depleted regions (NDRs). The precise positions of TSSs may form a focused pattern with one predominant TSS position, or a dispersed pattern, where TSSs can be detected within a broader sequence window that is characteristic of housekeeping genes[4]. A dispersed pattern of initiation maintains steady levels of transcription, yet results in RNA molecules that differ in their 5′ sequences, which may affect transcript stability[5] or alternative translation[6]. Additional transcript isoforms result from transcription on the antisense strand of genes to yield long non-coding RNA (antisense lncRNA)[7]. Antisense lncRNA transcription may regulate expression of the corresponding mRNA transcript[8]. Transcriptional initiation within genes (intragenic initiation) may result in mRNA isoforms encoding protein variants lacking N-terminal protein domains[9]. Chromatin-based mechanisms counter-acting intragenic TSSs during RNAPII elongation could affect the diversity of proteins derived from a gene[10]. However, it remains an open question whether intragenic TSSs usually connect to the canonical gene 3′-end, or if short transcript isoforms are generated that may not be translated.

Alternative transcriptional termination shapes isoform expression through differential selection of the 3′-end of transcripts[11,12]. Transcriptional termination at intragenic polyadenylation sites (PASs) results in mRNA isoforms lacking C-terminal protein domains and represents a mechanism regulating a wide range of biological processes[12]. In addition, the process of transcriptional termination is tightly connected to RNA surveillance pathways that determine RNA stability and nuclear export[13]. The nuclear exosome represents a 3′-to-5′ exonuclease that associates with RNAPII to degrade nascent RNAs[7]. Transcriptome analyses in nuclear exosome mutants thus reveal RNA molecules that are transcribed yet rapidly targeted for RNA degradation (i.e. cryptic transcripts), for example widespread divergent lncRNA transcription from gene promoter NDRs in the opposite direction respective to the mRNA[7]. Transcription in each direction of gene promoter NDRs is also selectively controlled at the level of initiation by chromatin-based mechanisms[14,15] and sequence-specific transcription factors[16]. Binding sites for the spliceosomal small nuclear RNA (snRNA) U1 selectively promote elongation of mRNA transcription over divergent lncRNA transcription[17]. The suppression of premature transcriptional termination (i.e. telescripting) by U1 is distinct from the splicing function of U1 (ref. [18]). pre-mRNA splicing represents an additional mechanism to amplify gene isoform diversity through alternative combinations of exons[19]. In summary, alternative pre-mRNA processing activities during transcriptional elongation contribute to the transcript isoforms generated from a single gene.

RNAPII density profiles across genes typically show two peaks: a first peak near the TSSs, and a second peak near PASs. The peak at PASs reflects a reduction of RNAPII movement that aids nascent transcript cleavage and polyadenylation by eukaryotic cleavage and polyadenylation specificity factors (CPSF) and cleavage stimulation factors (CstF)[11,20]. Peaks near TSSs represent RNAPII promoter-proximal pausing, where transcriptionally active RNAPII accumulates[21]. In metazoans, DRB sensitivity-inducing factor (DSIF) and negative elongation factor (NELF) stabilise RNAPII pausing[22]. RNAPII transcription elongation factors such as positive transcription elongation factor b (p-TEFb) and polymerase associate factor I (PAF-I) trigger disassociation of NELF and DSIF and promote transcriptional elongation[23,24]. The stability of paused RNAPII complexes in metazoans varies greatly between genes[25–27], supporting the idea that RNAPII turnover affects the accumulation of RNAPII complexes near TSSs[28]. RNAPII turnover at PASs as part of transcriptional termination coincides with peaks of RNAPII density[11]. As there is known RNAPII accumulation at PASs at the ends of genes, it is possible that the accumulation of RNAPII at the pausing sites is a result of the same process, but coupled to promoter-proximal polyadenylation. Indeed, short transcripts at genes with unstable promoter–promoter proximal RNAPII pausing are reported in Drosophila[29] that rely on transcriptional termination by the Integrator complex[30,31], that also functions in 3′-end processing of relatively short snRNA[32]. The prevalence of promoter-proximal transcriptional termination in additional organisms and potential roles of this phenomenon in gene expression remain an active research area.

Regulation of gene expression underpins key adjustments of sessile organisms (e.g. plants) to changing environments. In plants, a variable pattern of TSSs at gene promoters also contributes to heterogeneity of gene expression[33]. Although antisense lncRNA represent common gene isoforms, other gene isoforms such as divergent lncRNA are less frequent compared to metazoans[34]. Chromatin-based repression of intragenic RNAPII initiation relies on histone chaperone activity of the FACT complex. However, repression of intragenic TSSs during RNAPII elongation in plants is associated with histone 3 lysine 4 mono-methylation (H3K4me1) rather than histone 3 lysine 36 tri-methylation (H3K36me3), highlighting intriguing diversity in the use of chromatin-based signalling between organisms[35]. Plants and metazoans form equivalent CPSF/CstF protein complexes to mediate polyadenylation of mRNA[36,37] and integrator to mediate snRNA termination[38]. Similar to metazoans, nuclear exosome variants perform transcript surveillance[39,40]. An expansion of the RRM-domain RNA-binding gene family in plants specifies alternative polyadenylation sites[41] and regulates key aspects of plant development[42]. Interestingly, plant genomes lack NELF homologs yet display an accumulation of RNAPII near gene promoters[43]. A strong correlation between the position of the +1 nucleosome indicates a nucleosome-defined mechanism for promoter-proximal stalling in plants[34]. The functional significance of promoter-proximal stalling for gene isoform expression in plants is unclear.

Eukaryotic primary transcripts may extend up to a couple of megabases and provide extensive opportunities for co-transcriptional regulation of gene isoform diversity. Commonly used transcriptomics methods based on short-read sequencing detect either the TSS or the PAS of a given transcript. The connections between alternative TSSs to variable PASs that are key for gene isoform diversity thus remain poorly understood. Transcript Isoform sequencing (TIF-seq) brings TSS/PAS pairs of individual transcript molecules into close proximity during next-generation sequencing library construction to enable the identification of both RNA ends[44]. TIF-seq resolves transcript heterogeneity in budding yeast, representing a small eukaryotic genome with low average transcript length[45].

Here, we define TSS/PAS pairs for individual transcripts in the model plant Arabidopsis thaliana using an improved TIF-seq protocol suitable for larger genomes. TIF-seq data suggest on average over four isoforms corresponding to variable TSS/PAS pairs per expressed gene across different environmental

conditions. TIF-seq in *Arabidopsis* nuclear exosome mutants reveals the cryptic transcriptome, in particular short promoter-proximal RNAs (sppRNAs) from nascent transcript cleavage and polyadenylation shortly after initiation. The location of sppRNAs coincides with promoter-proximal RNAPII stalling. These data connect promoter-proximal RNAPII stalling and transcriptional termination in plants, with important implications for the regulation of gene expression in biotechnology.

## Results

**Mapping transcript isoform heterogeneity in *Arabidopsis*.** We mapped corresponding TSS/PAS pairs for individual RNA molecules genome-wide in *Arabidopsis thaliana* seedlings using an improved TIF-seq protocol[45] (Methods; Supplementary Figs. 1 and 2). We compared transcript boundaries detected by TIF-seq to boundaries determined by methods identifying TSSs[35] and PASs by Direct RNA sequencing (DRS)[46]. Overall, TIF-seq data correlated well to data mapping each transcript boundary separately (Supplementary Fig. 3a, b). We generated two biological repeat TIF-seq libraries for *Arabidopsis* Col-0 wild type that correlated well with each other ($r = 0.95$, Supplementary Fig. 3c). TIF-seq detected transcript boundaries for thousands of putative gene isoforms, such as those visible at the *GT-2 LIKE 1* (*GTL1*) gene (Fig. 1a). We combined transcript isoforms with 5′ and 3′ end sites co-occurring within 20 nt windows into clusters, and eliminated those with 5′ and 3′ mispriming, yielding 50,000 unique TIF-clusters (Methods; Supplementary Table 1). These analyses estimate an average of 4.3 distinct isoforms corresponding to alternative TSS/PAS pairs per expressed gene (Supplementary Fig. 3d). TIF-seq validated previously characterised alternative gene isoforms in *Arabidopsis* that result from alterative TSSs or PASs (Supplementary Fig. 3e, f), as well as antisense lncRNA variants (Supplementary Fig. 3g). Although most TSS/PAS pairs map annotated gene boundaries (Fig. 1b–d), considerable variability exists in 5′- and 3′-untranslated region (UTR) length (Fig. 1e). These variations might impact RNA stability, targeting and translation[47]. Overall, our TIF-seq data revealed thousands of alternative transcript isoforms encoded by the *Arabidopsis* genome.

**Chromatin-based control of intragenic initiation.** Our TIF-seq data highlight intragenic initiation of transcription that terminates at canonical gene ends as a common origin of plant isoform diversity (Fig. 1d). In *Arabidopsis*, intragenic TSSs are suppressed by a chromatin-based mechanism involving the conserved RNAPII-associated histone chaperone complex FACT, consisting of SPT16 and SSRP1 (ref. [35]). However, it remained unclear whether intragenic initiation repressed by FACT generates mRNA isoforms, or perhaps shorter variants. We performed TIF-seq in mutants of both *Arabidopsis* FACT subunits to resolve transcripts from intragenic initiation more clearly. Biological repeats of TIF-seq libraries for *spt16-1* ($r = 0.99$) and *ssrp1-2* ($r = 0.97$) showed high correlation (Supplementary Fig. 4a, b). Similar to wild type, we detected about four isoforms per gene on average in *fact* mutants (Supplementary Fig. 4c, d). Moreover, the categories of transcript isoforms were largely similar to wild type, with internal initiation as a key source for transcript isoform diversity in *fact* mutants (Fig. 2a, b). Genome browser screenshots of genes containing FACT-repressed intragenic TSSs[35] supported the idea that intragenic TSSs often extended to canonical PASs (Fig. 2c, d). A focused analysis of FACT-repressed intragenic TSSs genome-wide revealed that intragenic TSSs produce alternative gene isoforms that terminate at gene ends (Fig. 2e, f). Thus, intragenic initiation regulated by FACT connects with canonical PASs in *Arabidopsis* to yield alternative mRNA isoforms.

**Mapping cryptic transcript isoforms reveal sppRNAs.** Transcriptome analyses in nuclear exosome mutants facilitate the detection of many cryptic RNA species[7]. *Arabidopsis* mutants lacking the RNA helicase HUA ENHANCER 2 (HEN2) display defective exosome activity[40]. We performed TIF-seq in the *hen2-2* mutant to map cryptic transcript isoforms in *Arabidopsis*. Biological repeats of *hen2-2* TIF-seq libraries correlated well ($r = 0.99$, Supplementary Fig. 5a), yet we could not detect an increase in transcript isoforms per gene compared to wild type (Supplementary Fig. 5b). TIF-seq data of *hen2-2* mutants revealed an increased proportion of intergenic transcripts and antisense isoforms[39] (Fig. 3a, b). Interestingly, we discovered a high proportion of internal termination, representing prematurely terminated transcripts (Fig. 3b). In particular, TIF-seq data in *hen2-2* mutants resolved a population of transcripts with 3′-ends in close proximity to promoter TSSs (Fig. 3c). Size distribution plots of transcript isoforms show that short (<200 nt) transcripts accumulate in *hen2-2* that mostly terminate close to their respective promoter TSSs (Fig. 3d). We defined this transcript class as short promoter-proximal RNAs (sppRNAs). We determined a median sppRNA length of ~93 nt (Fig. 3e). In total, we detected sppRNAs at 13–15% of expressed genes (Supplementary Table 1). Genes with sppRNAs show elevated nascent RNAPII transcription compared to genes without detectable sppRNAs (Fig. 3f), indicating that sppRNAs are detected at genes with generally higher RNAPII transcriptional activity. Collectively, TIF-seq revealed sppRNAs as a prevalent feature of plant gene expression.

Genome browser screenshots of TIF-seq data displayed prominent *hen2-2*-specific sppRNAs, for example at the *MAP KINASE 20* (*MPK20*) locus (Fig. 3g, Supplementary Fig. 5c–f). We validated sppRNAs by RT-qPCR at five genes (Fig. 3h; and Supplementary Fig. 6). Although sppRNAs at additional genes also generally increased in *hen2-2*, low sppRNA levels were detected above background in wild type by RT-qPCR (Supplementary Fig. 6). Northern blotting revealed a broad size distribution for *MPK20* sppRNAs in *hen2-2*, consistent with alternative sppRNA PASs (Fig. 3i). We detected sppRNA specifically with probes against the 5′-end of the *MKP20* gene, confirming that sppRNA originate from the beginning of transcription units (Fig. 3i). We next defined the *MPK20* sppRNA size profile at higher resolution using polyacrylamide gel electrophoresis (PAGE) followed by northern blotting (Fig. 3j). These data estimated the size range of *MPK20* sppRNA to 55–100 nts, consistent with our genome-wide estimate of the median length (Fig. 3e, j). Interestingly, sppRNA accumulation in *hen2-2* did not significantly alter mRNA levels for tested genes (Supplementary Fig. 7, Fig. 3i, j). These data suggest that sppRNA stabilisation mediates no statistically significant effect on cognate mRNA levels. Together, these results show that premature termination near *Arabidopsis* promoters generates relatively short-lived capped and polyadenylated transcripts that are targeted by the nuclear exosome RNA degradation pathway.

**sppRNA formation is linked to gene expression.** To query plant transcript isoform diversity in different environmental conditions we performed TIF-seq in wild type and *hen2-2* seedlings following 3 h of 4 °C treatment, which is sufficient to trigger transcriptional responses that promote cold acclimation[8]. Biological repeats of TIF-seq libraries at 3 h of 4 °C for wild type ($r = 0.8$) and *hen2-2* ($r = 0.99$) were correlated, indicated reproducibility between repeats (Supplementary Fig. 8a, b). Compared to ambient temperature, we detected an equal number of isoforms per expressed gene in the cold (Supplementary Fig. 8c, d) and a largely identical distribution of

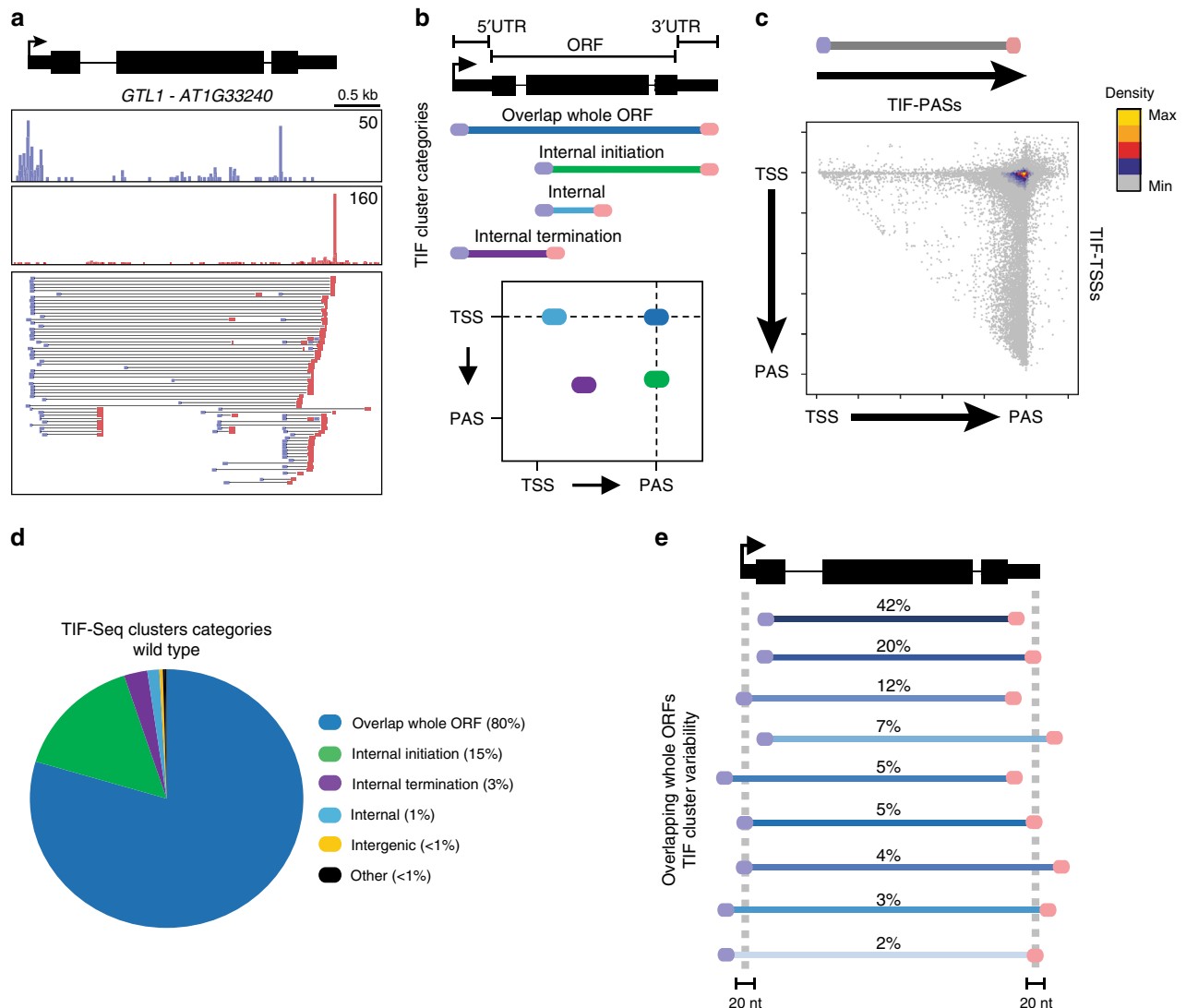

**Fig. 1 Mapping TU boundaries in *Arabidopsis* by TIF-seq. a** Genome browser screenshot of TSSs (TSS-seq)[35], PASs (Direct RNA-seq; DR-seq)[73], TSS/PAS pairs (TIF-seq) and TIF-clusters at the *GTL1* gene. **b** Schematic of TIF-TSS/PAS pairs representing different TIF-cluster categories and 2D illustration of these positions (blue circles are TSSs and red circles PASs) with respect to genome annotations. Each line and its colour correspond to the coloured circles in the schematic scatterplot. **c** Scatterplot of TIF-cluster TSS/PAS pairs in wild type. Each point represents one TIF-seq cluster and its TIF-TSS and PAS distance to respective annotated gene boundaries and normalised to gene length. The colour scale represents 2D position enrichment density of clusters from low (grey) to high (yellow). **d** Pie chart of TIF-cluster categories proportions present in wild type. **e** Schematic representation and percentage of subcategories for TIF-clusters overlapping whole ORFs. Source data of **d** and **e** are provided in the Source Data file.

transcript isoform classes (Fig. 4a, b). A comparison between isoforms detected in *hen2-2* and wild type during the cold confirmed the enrichment of internal termination (Supplementary Fig. 8e). TIF-seq validated the cold-induced isoforms of the lncRNA *SVALKA* (Supplementary Fig. 8f). A scatterplot of TSS/PAS pairs of *hen2-2* TIF-seq data during the cold revealed a high density of transcript ends adjacent to TSSs compared to wild-type data in the cold (Fig. 4c, d). A comparison of transcript size distribution revealed an increase in short transcripts in *hen2-2* compared to wild type in the cold (Fig. 4e). A focused scatterplot analysis revealed that these short (<200 nt) transcript isoforms mapped near TSSs, matching the location of sppRNA identified in ambient temperature (Fig. 4f). In conclusion, we found similarities between transcript isoform distribution in ambient temperature and during the initial plant responses to cold temperature. sppRNA detection in *hen2-2* in the cold suggests that sppRNA represent common plant gene isoforms across environmental conditions.

Our results support the idea that transcription events initiating at the same TSS can form full-length mRNAs or alternatively sppRNAs. This could represent a mechanism to adjust gene expression by selective transcriptional termination[11]. Cold temperature affects the efficiency of transcriptional termination in plants[34], and Gene Ontology term (GO-term) analyses revealed that a significant number of genes with sppRNAs in ambient temperature are cold-regulated (Supplementary Data 1). To test the idea that cold temperature may regulate gene expression through sppRNA formation, we compared wild-type TSS-seq data that detect the mRNA of sppRNA genes in the cold and ambient temperature. Genes with sppRNA isoforms may reduce sppRNA formation and increase mRNA formation in the cold, which would represent a pattern of transcripts consistent with gene regulation by selective transcription termination (Supplementary Fig. 9a). We detected a profile consistent with mRNA increases at the expense of sppRNAs for some (38 of 1153) genes following cold treatment (Supplementary Fig. 9b).

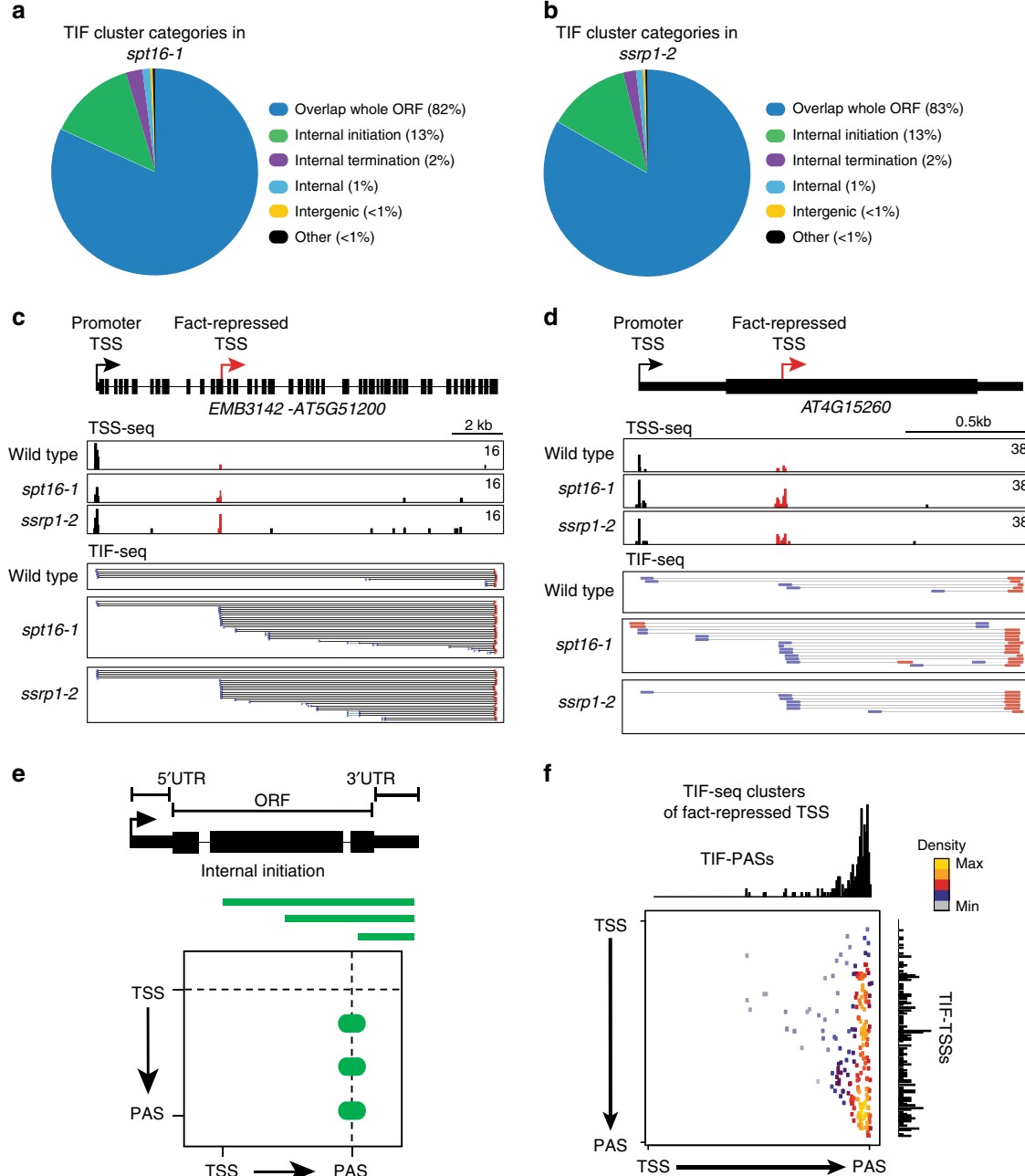

**Fig. 2 The FACT complex represses initiation of intragenic isoforms. a** Pie chart of TIF-cluster categories proportions present in *spt16-1*. **b** Pie chart of TIF-cluster categories proportions present in *ssrp1-2*. **c, d** Genome browser screenshots of the genes *AT5G51200* (**c**) and *AT4G15260* (**d**). TSS-seq data (top) for wild type, *spt16-1* and *ssrp1-2*, with fact-repressed intragenic TSS in red. TSS/PAS pairs by TIF-seq (below) are indicated as lines connecting TSSs (blue) and PASs (orange). Inverted TIF-cluster orientation for *spt16-1* in **d** indicated antisense lncRNA isoforms. **e** Schematic representation of illustrating profile of TIF-TSS/PAS pairs representing internal initiation. TIF-cluster and 2D illustration of their positions with respect to genome annotations are indicated in green lines and circles. **f** Scatterplot representing the TSS/PAS pair distance to annotated TSS/PAS for a subset of TIF-clusters from the combined *spt16-1* and *ssrp1-2* TIF-seq data sets. The selected data strictly overlap the published *fact*-specific TSSs genomic positions identified through TSS-seq differential expression analysis in both *spt16-1* and *ssrp1-2* mutants. The colour scale represents 2D position enrichment density of clusters from low (grey) to high (yellow). Source data of **a, b** are provided in the Source Data file.

These data indicate that selective termination to yield sppRNA may modestly contribute to transcriptome adaptations in this condition. However, most genes with sppRNA show no evidence for gene regulation by selective termination, and an equivalent fraction of mRNA without sppRNAs is cold-induced (Supplementary Fig. 9b). Instead, we detect sppRNAs at many cold-induced genes following activation of gene expression in the cold

(Supplementary Fig. 9c, d), consistent with sppRNA formation linked to increased RNAPII activity (Fig. 3f). In summary, we detected a positive correlation between sppRNA formation and plant gene expression across temperatures.

To test the potential roles of sppRNA in transcriptional activation we assayed the effects of sppRNA located in two 5′-UTRs on reporter gene activity. The 5′-UTRs of the sppRNA

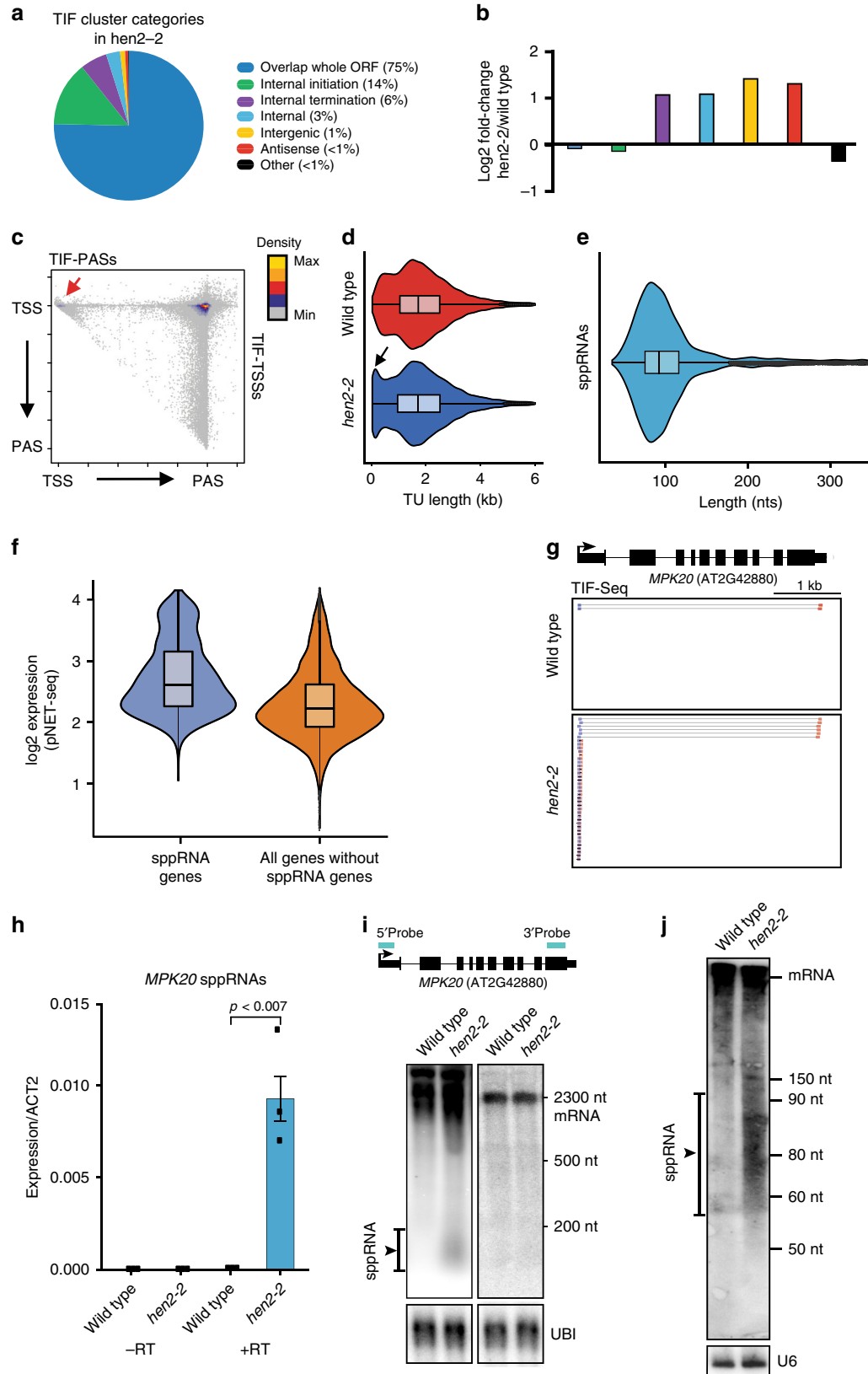

genes *MPK20* and *ACT8* contain prominent sppRNA. We quantified Yellow Fluorescent Protein (YFP) fluorescence following transient expression in *N. benthamiana* leaves in constructs with wild-type 5′-UTR sequence and mutant versions carrying deletions of the sppRNA sequences. Constructs carrying deletions

of sppRNA sequences (ΔsppRNA) visually reduced YFP fluorescence compared to the wild-type 5′-UTRs (Supplementary Fig. 10a–d). Quantification of YFP fluorescence revealed a statistically significant reduction in both constructs carrying sppRNA deletions (Supplementary Fig. 10e). These data are

**Fig. 3 TIF-seq in hen2-2 reveals promoter-proximal termination. a** Pie chart of TIF-cluster categories proportions in *hen2-2* mutants. **b** Log$_2$ fold-change of TIF-cluster categories in *hen2-2* compared to wild type. Colour code as indicated in **a**. **c** Scatterplot of TIF-cluster TSS/PAS pairs in *hen2-2*. Red arrow indicates TSS/PAS pairs within initiation regions. The colour scale represents 2D position enrichment density of clusters from low (grey) to high (yellow). **d** TIF-cluster sizes in wild type (*n* = 41,190) and *hen2-2* (*n* = 48,225). Black arrow indicates enrichment of short TUs in *hen2-2*. **e** Distribution of short TIF-clusters sizes (<350 nt). sppRNAs have a median length of 93 nt. **f** The distribution of nascent gene body RNAPII transcriptional activity for genes with sppRNAs compared to all other genes from pNET-seq (8WG16) in wild-type *Arabidopsis*. For both classes tested: *n* = 1639. Gene body is defined as +200 nts from annotated TSS and −200 nts from annotated PAS. The boxes in **d**–**f** bounds the iQR divided by the median, and whiskers extend to a maximum of 1.5× iQR beyond the box. **g** Genome browser screenshot of full-length *MPK20* mRNA and sppRNAs revealed in the *hen2-2* track. **h** RT-qPCR for *MPK20* sppRNA levels in wild type and *hen2-2*. With reverse transcriptase (+RT) and without (−RT). Data are presented as mean ± SEM from three independent experiments. Black dots indicate individual data points (*n* = 3). *p*-value between wild type and *hen2-2* are derived from two-sided Student's *t*-test. **i** Profiling of *MPK20* sppRNAs by agarose gel northern blotting in wild type and *hen2-2*. *UBI* is a loading control. Blot picture is representative of one out of three biological replicates. TIF-clusters in *hen2-2* and positions of northern probes for sppRNA (5′ probe) and control probe for mRNA (3′ Probe) are indicated. **j** PAGE northern blot in wild type and *hen2-2*. Loading control is the U6 snRNA. The blot picture is representative of one out of three biological replicates. Enrichment of sppRNA is indicated in **i**–**j** by a black line. Source data of **a**, **b**, **h** and **i**, **j** are provided in the Source Data file.

consistent with roles for sppRNA in activating gene expression and support the positive correlation between sppRNA detection and gene expression.

**Molecular characteristics of sppRNA formation.** We predicted that sppRNAs result from promoter-proximal RNAPII regulation. To examine influence of nascent RNAPII activity on sppRNA formation, we plotted plant Native Elongating Transcript sequencing (pNET-seq) data for genes with detectable sppRNAs[43]. The end positions of sppRNAs (i.e. PAS) map within peaks of RNAPII occupancy (Fig. 5a, b). We also observed this pattern for pNET-seq data obtained with a different RNAPII antibody (Supplementary Fig. 11)[43]. These data suggested a link between RNAPII promoter-proximal stalling and sppRNA formation. Indeed, RNAPII stalling was specifically enriched near promoters for genes with sppRNAs when compared against a control set of genes without detectable sppRNAs but with otherwise equal gene body RNAPII activity (Fig. 5c). Overall, significantly increased RNAPII stalling shortly after initiation correlates strongly with promoter-proximal cleavage and polyadenylation generating sppRNAs (Supplementary Fig. 12).

Promoter sequence elements affect promoter-proximal RNA-PII stalling in metazoans[48]. Our analyses of promoter elements revealed an enrichment of two motifs for sppRNA genes (Supplementary Fig. 13a, b). We detected the GAGA-box motif at sppRNA genes around the TSS (Supplementary Fig. 13c, d). A comparative plot between sppRNA genes and the control set with equivalent distribution of RNAPII transcription clearly visualises the GAGA-box motif enrichment (Fig. 5d, e). Although the GAGA-box motif enrichment is particularly clear downstream of the TSS, the enrichment zone extends about 10-nt upstream of the TSS. Interestingly, the GAGA-box motif and promoter-proximal pausing are connected in metazoans[48], where the GAGA-box motif is positioned further upstream of the TSS.

We next used DREME[49] and RSAT[50] to test for statistically enriched promoter sequence motifs in sppRNA genes. Both methods identified an enrichment of the HTGGGCY or RGCCCAW reverse complement motif at sppRNA genes (Supplementary Fig. 13e, f). A comparative motif plot between sppRNA genes and the control set with equal RNAPII transcription strikingly visualises motif enrichment (Fig. 5f, g). Interestingly, the HTGGGCY motif is associated with binding of TCP transcription factors[51] that regulate cell division and plant development[52]. In conclusion, our promoter *cis*-element analyses identified TCP transcription factor binding sites upstream of the TSS of sppRNA genes, and GAGA factor motifs predominantly downstream of the TSS. We find that DNA sequence elements and promoter-proximal RNAPII stalling distinguish sppRNA genes and may contribute to sppRNA formation near plant gene TSSs.

**Promoter-proximal RNAPII termination generates sppRNA.** Premature transcriptional termination of RNAPII transcription shortly after transcriptional initiation would reconcile the position of sppRNA PASs. The coincidence of sppRNA formation and promoter-proximal RNAPII stalling suggested co-transcriptional regulation by transcription elongation factors (EFs). p-TEFb phosphorylates RNAPII and associated factors to drive elongation[24]. RNAPII peaks at sppRNA genes are particularly sensitive to treatment with the p-TEFb inhibitor Flavopiridol (Supplementary Fig. 14), linking stalled RNAPII to sppRNA formation. To test the connection between RNAPII stalling, p-TEFb, and premature termination, we used an *Arabidopsis* mutant for the p-TEFb kinase CYCLIN-DEPENDENT KINASE C2 (CDKC2): *cdkc2-2* (ref. [53]). RT-qPCR analyses of sppRNA generated from five loci revealed that sppRNA levels increased in *cdkc2-2* compared to wild type (Fig. 6a). Notably, sppRNA production increased in *vip5* (Fig. 6a), a mutant of the *VERNALIZATION INDEPENDENCE 5* (*VIP5*) gene[54], which encodes a conserved subunit of the PAF-I complex. These data are consistent with RNAPII elongation repressing sppRNA formation. Thus, EF activity may help to bypass a gene expression checkpoint associated with sppRNA formation.

Transcript isoform detection by TIF-seq relies on 3′-terminal poly(A) tails that are associated with the activities of eukaryotic CPSF/CstF 3′-end formation complexes. To test the contribution of *Arabidopsis* Cleavage Stimulation Factor 64 (CstF64) on sppRNA 3′-end formation we assayed sppRNA and mRNA in the *cstf64-2* mutant[42]. Even though sppRNA are polyadenylated, we could not resolve decreased sppRNA in *ctsf64-2* compared to wild type (Fig. 6a). However, we detected a decreased sppRNA/mRNA ratio in *ctsf64-2* compared to wild type (Supplementary Fig. 15a). These data suggested that the CstF/CPSF complex may promote promoter-proximal polyadenylation of sppRNA on the expense of full-length mRNA formation. To test this further, we visually inspection of Poly-(A) Test Sequencing (PAT-seq) data in Cleavage Stimulation Factor 77 (CstF77) mutants compared to wild type. At loci such as the *At4g29810* gene, PAT-seq revealed relative changes in poly-(A) site usage consistent with usage of distal sites (Fig. 6b). Perhaps counter-intuitively, *Arabidopsis* CPSF/CstF mutants, that are defective in 3′-end formation, could show increased mRNA formation at sppRNA genes. Our RT-qPCR analyses of mRNA expression for the panel of five sppRNA genes supported this idea for some targets (Supplementary Fig. 15b). To test whether mRNA expression of sppRNA genes increases in *cstf/cpsf* mutants genome-wide, we analysed PAT-seq data in mutants of *Arabidopsis* CstF77 (ref. [55]) and

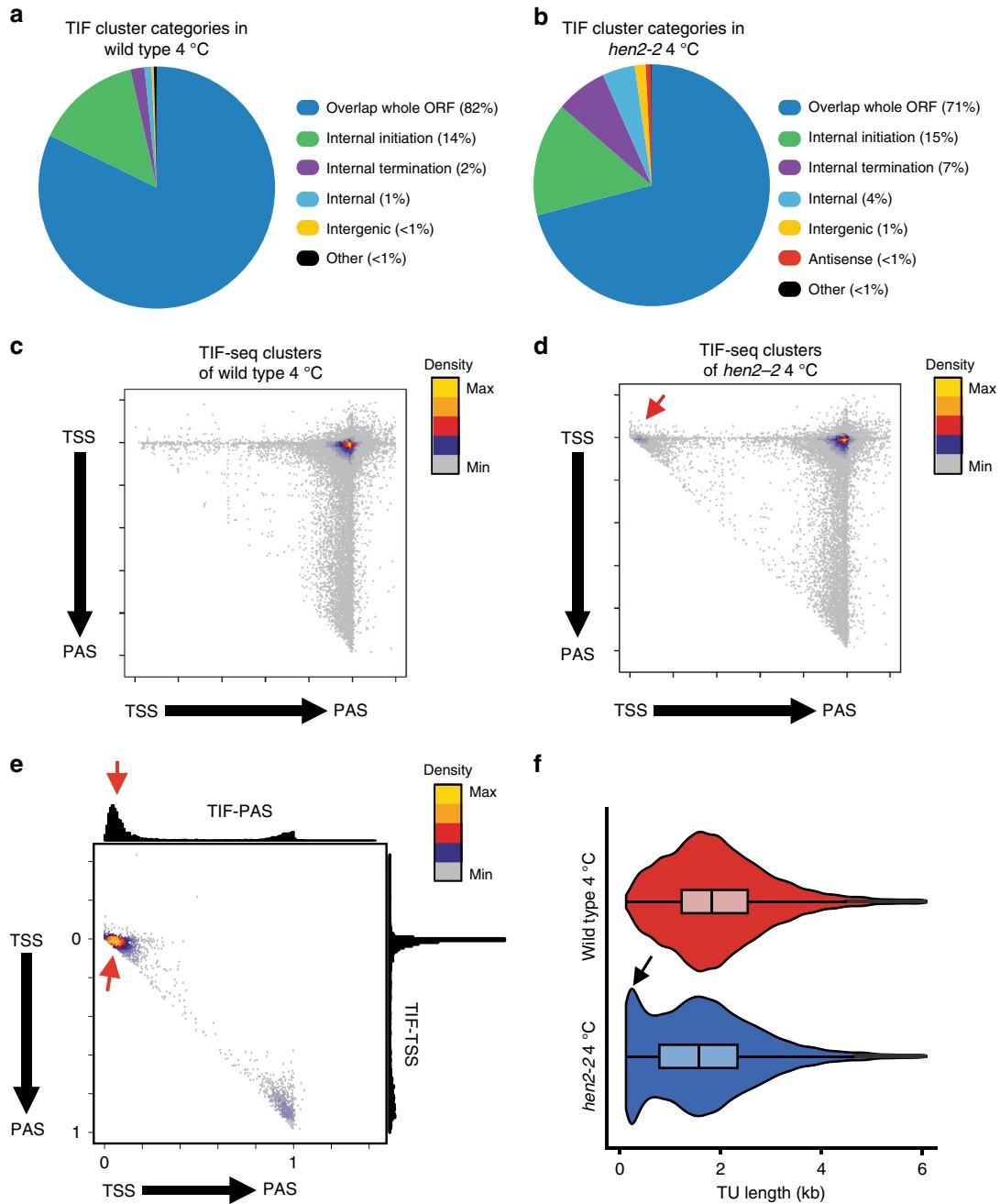

**Fig. 4 Transcript isoform mapping during the response to cold temperature. a** Pie chart of TIF-cluster categories in 4 °C-treated wild type. **b** Pie chart of TIF-cluster categories in 4 °C-treated *hen2-2*. **c** Scatterplot of TIF-cluster TSS/PAS pairs in 4 °C-treated wild type. **d** Scatterplot of TIF-cluster TSS/PAS pairs in 4 °C-treated *hen2-2*. **e** Scatterplot of <300 bp TIF-cluster TSS/PAS pairs in *hen2-2*. Histograms and bar plots display TIF-TSSs/-PASs distribution normalised to gene length from annotated TSS to PAS. The colour scale in **c**–**e** represents 2D position enrichment density of clusters from low (grey) to high (yellow). **f** TIF-cluster sizes in 4 °C-treated wild type and *hen2-2*. Black arrow indicates enrichment of short transcripts in 4 °C-treated *hen2-2*, the box bounds the iQR divided by the median, and whiskers extend to a maximum of 1.5× iQR beyond the box. Source data of **a**, **b** are provided in the Source Data file.

CPSF100 (ref. [56]). We detected higher mRNA expression at sppRNA genes compared to the control set specific to the two *cstf/cpsf* mutants (Fig. 6c, d). These data offer genome-wide support for a model where promoter-proximal RNAPII stalling and sppRNA formation by CPSF/CstF may limit mRNA formation. The Integrator complex mediates promoter-proximal 3′-end formation in metazoans[30]. To test the role of Integrator in sppRNA formation, we performed RT-qPCR analyses in the *Arabidopsis* Integrator mutants, *dsp1-1* and *dsp4-1*, disrupting the *DEFECTIVE in snRNA Processing 1* and *4* (*DSP1 and DSP4*)

genes, respectively[38]. We detected decreased sppRNA formation, suggesting that the *Arabidopsis* Integrator complex promotes sppRNA 3′-end formation (Fig. 6a). At four of five genes assayed by RT-qPCR, we detected a decreased mRNA/sppRNA ratio in *Arabidopsis* integrator mutants (Supplementary Fig. 15a). Our data suggest that integrator participates in sppRNA formation. Collectively, our findings indicate sppRNA formation associated with promoter-proximal RNAPII stalling as checkpoint for plant gene expression (Fig. 6e). Although sppRNA are generally associated with high transcriptional activity, RNAPII elongation

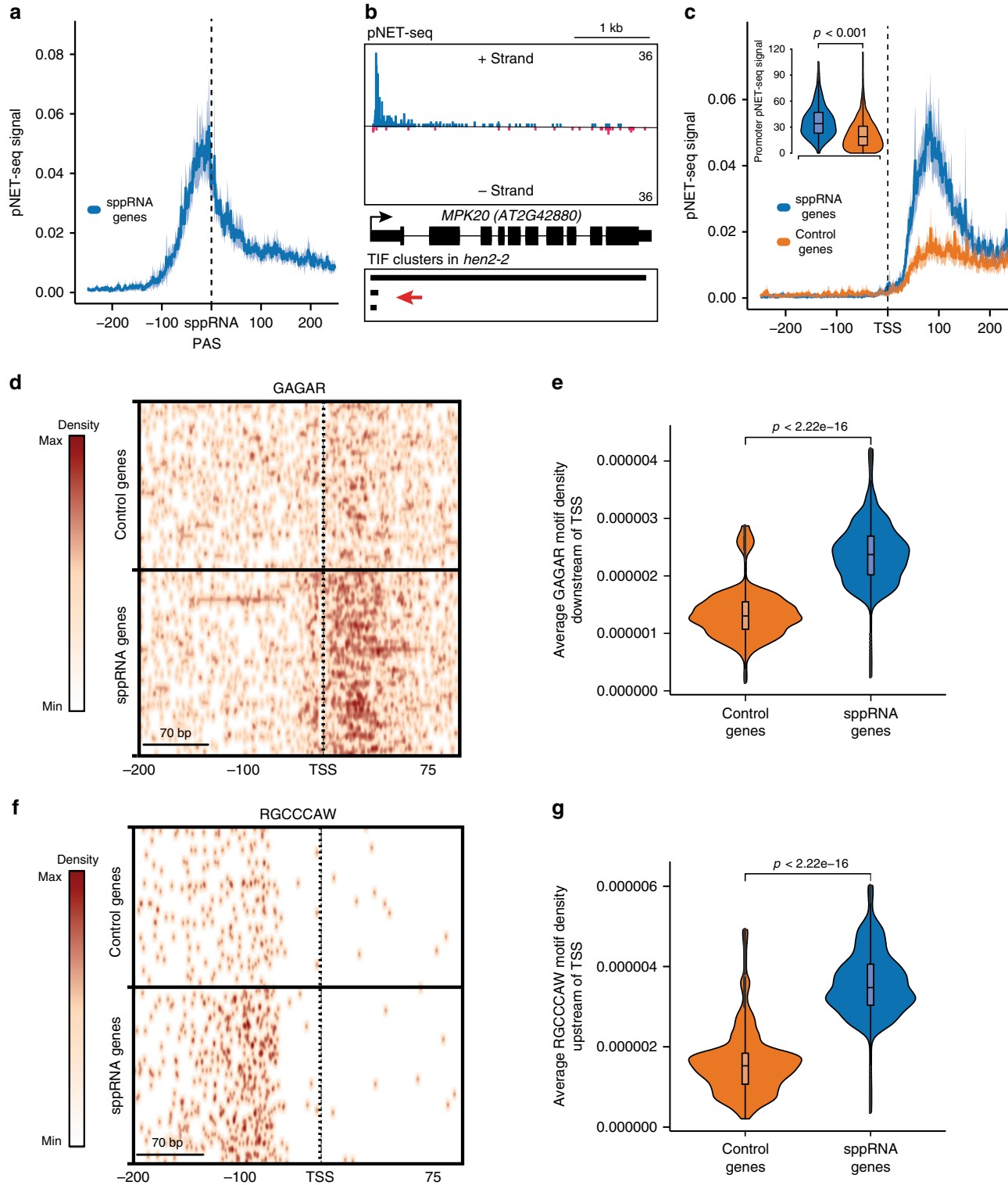

factors may balance bursts of productive full-length isoform production with promoter-proximal termination events to control plant gene expression.

## Discussion

Our improved TIF-seq protocol for genomes larger than budding yeast[45] revealed transcript isoform diversity of four genotypes at ambient temperature and two genotypes during cold treatment, totalling 12 TIF-seq libraries. Remarkably, we identified four

transcript isoforms per expressed gene on average, insensitive to temperature variations, defective chromatin assembly and defective nuclear RNA degradation. A strength of TIF-seq is the ability to capture transcript isoform diversity arising through heterogeneous transcript ends. We note that TIF-seq would miss isoform diversity that arises in the middle of transcripts, for example through alternative pre-mRNA splicing[19]. Moreover, transcripts carrying epitranscriptome variations such as alternative 5′-cap structures[57] may be incompatible with our sequencing library strategy. The combination of TIF-seq complemented by long read

**Fig. 5 RNAPII stalling coincides with regulated termination near Arabidopsis promoters. a** Wild-type pNET-seq (8WG16)[43] coverage averaged around sppRNA PAS genomic positions for sppRNAs genes (*n* = 1155). **b** Genome browser screenshot of wild-type pNET-seq reads at the *MPK20* gene. TIF-cluster diagram indicates annotates mRNA and sppRNA transcript isoforms. Red arrow indicates sppRNAs TIF-cluster. **c** Wild-type pNET-seq (8WG16)[43] coverage averaged around sppRNA PAS genomic positions for sppRNAs genes (*n* = 1155) and a control set of genes displaying equal distribution of nascent gene body transcription (*n* = 1153). Inset violin plot depicts significantly increased average pNET-seq signal within first 150 nt from TSS for genes with sppRNAs (Wilcoxon test: *p* < 2.2e−16) compared to the control genes. Shading in **a** and **c** represents the calculated confidence interval on the mean of 95%. **d** Heatmap of *cis*-element GAGA-box (GAGAR) in genomic regions 200 bp upstream and 150 bp downstream of the TSS of control genes with equal pNET-seq nascent transcription distribution to sppRNA genes (*n* = 1155). **e** Violin plot of the average GAGAR density downstream of the TSS (two-sided Wilcoxon test: *p* < 2.2e−16). **f** Heatmap of *cis*-element TCP-binding motif (RGCCCAW) in genomic regions 200 bp upstream and 150 bp downstream of the TSS of control genes with equal pNET-seq nascent transcription distribution to sppRNA genes (*n* = 1155). The colour scale in **d** and **f** represents motif frequency from low (white) to high (red). **g** Violin plot of the average GAGAR density upstream of the TSS (two-sided Wilcoxon test: *p* < 2.2e−16). The inset box in **c** and boxes in **e** and **g** bounds the iQR divided by the median, and whiskers extend to a maximum of 1.5× iQR beyond the box.

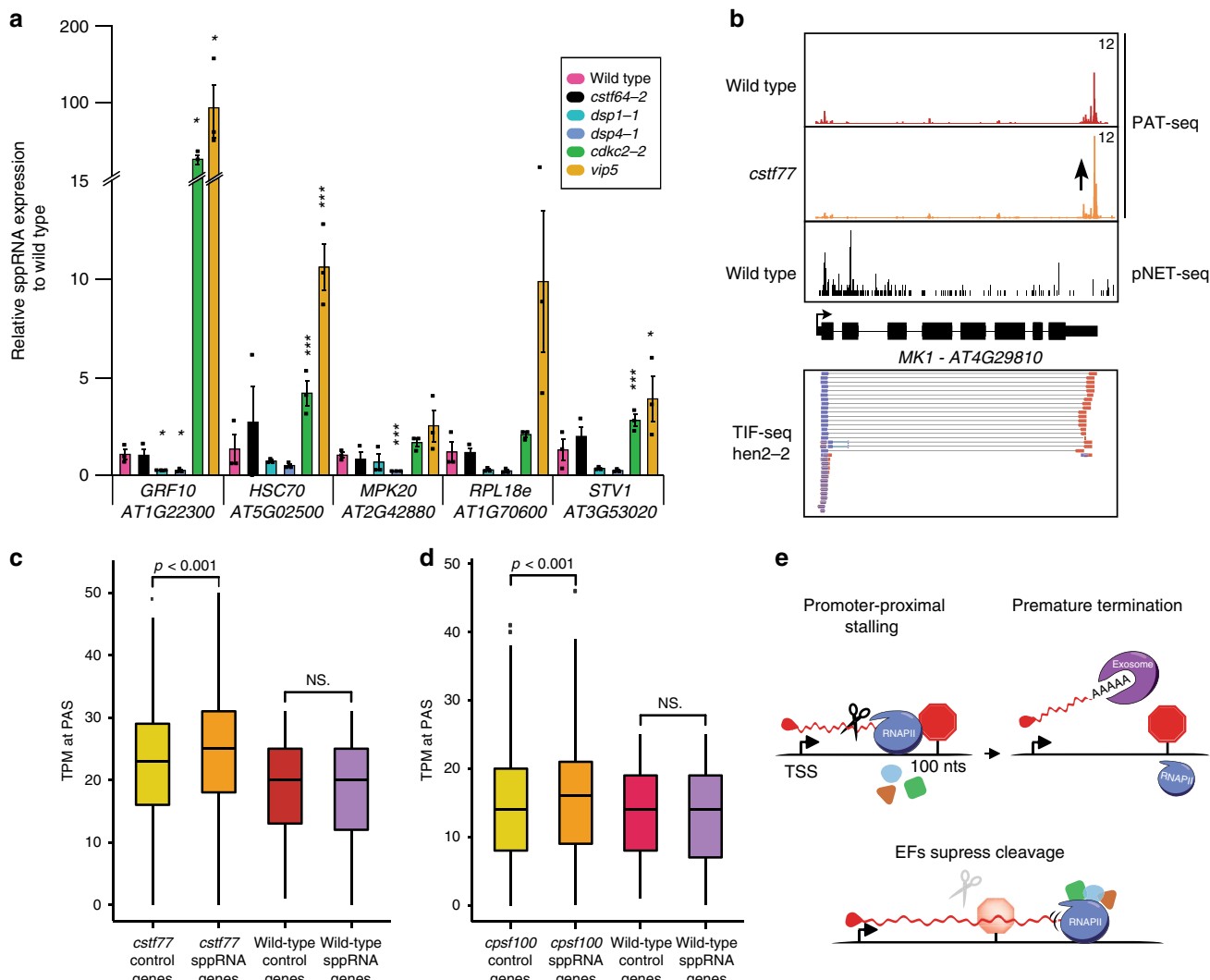

**Fig. 6 RNAPII elongation and termination shape transcript isoforms. a** RT-qPCR analyses of sppRNA for five genes with sppRNAs in wild type, *cdkc2-2*, *vip5*, *dsp1-1,dsp4-1* and *cstf64-2*. Data are presented as mean values ± SEM from three independent experiments. Black circles indicate individual data points (*n* = 3). Single asterisk denotes *p* < 0.05, two asterisks denote *p* < 0.01, whereas three asterisks denote *p* < 0.001 between mutant and wild type by two-sided Student's *t*-test. **b** Genome browser screenshot of the *MK1* gene. Visualisation of PAT-seq data for wild type (red) and *cstf77* (orange), pNET-seq data (black) and *hen2-2* TIF-seq data. Black arrow represents the increase of PAT-Seq signal in *cstf77*. **c** Distribution of TPM at mRNA PAS for sppRNA genes (*n* = 1173) and its specific control set of genes (*n* = 1170) (Methods) in wild type and *cstf77* mutant. **d** Distribution of TPM at mRNA PAS for sppRNA genes (*n* = 1161) and its specific control set of genes (*n* = 1129) (Method) in wild type and *cpsf100* mutant. In **c** and **d**, *p* < 0.001 denote the *p*-value calculated by two-sided Wilcoxon test between mRNA PAS usage for sppRNA genes compared to a set of control genes displaying equal distribution of nascent gene body transcription in the mutants, NS denotes no statistically significant difference. In **c** and **d**, the box bounds the iQR divided by the median, and whiskers extend to a maximum of 1.5× iQR beyond the box. **e** Schematic representation of promoter-proximal RNAPII stalling ~100 nt downstream of TSSs coinciding with nascent transcript cleavage, polyadenylation, and exosome-mediated degradation of sppRNAs. Elongation factors (EF) suppress promoter-proximal termination. Source data of **a** are provided in the Source Data file.

RNA sequencing methods appears suitable to gain powerful insight into transcript isoform diversity[58]. The synergies offered by TIF-seq and long read sequencing would enhance transcriptome annotations in non-model species beyond RNA-seq, that currently represents the starting point for investigation, for example in many plant species[59].

Our data inform the debate of what we should consider a gene[2]. The largest fraction of transcript variants detected by TIF-seq contained the full ORF, yet displayed variable 5′-UTR or 3′-UTR lengths. In metazoans, 3′-UTRs regulate protein function by several mechanisms[60]. The differences in 3′-UTRs we detected by TIF-seq could thus reflect a largely unexplored layer of regulation. Likewise, variability in 5′-UTRs can affect RNA structure, which may regulate gene activity through effects on translation[6]. We note that expression strategies for heterologous expression in biotechnology often lack endogenous UTR variations, potentially limiting beneficial properties for expression[61]. Our transcriptome annotation by TIF-seq provides the platform for future studies characterising the functional significance of variations in UTRs that distinguish plant transcript isoforms.

We detected intragenic transcription initiation as second largest source of isoform diversity. This type of isoform variation would encode a protein lacking N-terminal domains that often determine intracellular localisation[3]. TIF-seq clarified that transcripts initiating within gene bodies usually extend to the predicted gene PAS. However, the contributions of intragenic initiation to plant proteome diversity remain unresolved, and these transcript isoforms may be targets for cellular RNA degradation pathways[62].

TSS-seq data previously suggested that the *Arabidopsis* FACT complex represses intragenic TSSs[35]. TIF-seq data showed that FACT-repressed intragenic TSSs often extend to the canonical PASs. Repression of intragenic initiation of RNAPII transcription by FACT thus results in transcript isoforms that potentially encode proteins lacking N-terminal domains. These data support chromatin-based regulation of the transcript isoform diversity derived from a single gene. FACT represses intragenic TSSs also in yeast and metazoans[63,64], supporting the idea that our results may help to understand gene isoform regulation by FACT in additional organisms. FACT mutants affect plant growth and development[65,66], perhaps reflecting that regulation of intragenic initiation by FACT assists cellular decisions needed to execute plant developmental programmes.

TIF-seq data revealed alternative 3′-end formation sites close to TSSs that identified sppRNA. We note that most sppRNA represent cryptic transcripts, since sppRNA accumulate in nuclear exosome mutants (i.e. *hen2-2*). However, for some genes we detected sppRNA in TIF-seq libraries generated from wild type. sppRNAs co-localise with promoter-proximal stalled RNAPII complexes. The function of stalled RNAPII complexes is unclear; they may prevent new acts of initiation[67] and limit full-length isoform production by regulated pause-release mechanisms[25]. In mammals, RNAPII pauses shortly after the TSSs, and then stalls again further downstream near the position of the +1 nucleosome[68]. Notably, the extent to which RNAPII accumulation near metazoan TSSs represents transient transcriptional pausing or dynamic RNAPII turnover, perhaps through transcriptional termination, is actively debated[69].

Nucleosome-defined RNAPII stalling is dominant in *Arabidopsis*[34], consistent with the conspicuous absence of metazoan pausing factors in plant genomes. The sppRNA biogenesis mechanism thus likely differs from RNA species detected near initiation regions that may depend on metazoan-specific pausing factors[12,29]. sppRNA carry poly-(A) tails, our data thus support a biogenesis mechanism involving CstF/CPSF complexes. However, the Integrator complex assists sppRNA formation. The Integrator complex terminates transcription at unstable promoter-proximal pausing sites in metazoans[30]. Interestingly, our *cis*-element analyses identified the GAGA motif at sppRNA genes in plants. This motif affects metazoan RNAPII promoter-proximal pausing; albeit at a position further upstream[48]. In conclusion, our data identifies promoter-proximal termination of RNAPII transcription in plants that bears some tantalising parallels to a perhaps equivalent process in metazoans.

sppRNA detection supports transcriptional termination as the outcome of RNAPII stalling near the +1 nucleosome at many plant promoters. We tested for gene regulation by selective sppRNA formation at the expense of mRNA formation upon cold perception. Although these analyses were consistent with gene regulation at some selected loci, sppRNA formation largely correlated positively with RNAPII transcription and full-length gene isoform formation. Consistently, targeted deletion of sppRNA sequences in the 5′-UTR reduced reporter gene activity. On the other hand, our analyses of sppRNA/mRNA ratios in mutants defective in 3′-end formation pathways argue that sppRNA formation may limit full-length mRNA expression. Promoter-proximal RNAPII transcriptional termination resulting in sppRNA and associated RNAPII turnover may positively contribute to plant gene expression, perhaps by maintaining NDRs near TSSs to facilitate new rounds of transcription initiation. On the other hand, sppRNA formation reduces mRNA formation by limiting progression of a fraction of initiated RNAPII complexes. Although this reflects negative regulation of full-length mRNA isoforms, perhaps RNAPII complexes that fail quality control checkpoints trigger sppRNA formation to promote the quality of full-length mRNA expression. Elucidating the potential roles of sppRNA in gene regulation will remain a topic for future studies.

Our data are consistent with the idea that sppRNA result from stalled RNAPII complexes failing to enter productive elongation, perhaps representing a checkpoint for plant RNAPII transcription during early transcriptional elongation. In summary, our analyses of isoform diversity in plants inform on the debate about peaks of RNAPII occupancy shortly after TSSs, suggesting they often represent RNAPII complexes engaged in premature transcriptional termination.

## Methods

**Plant growth.** All *A. thaliana* lines used in this study are listed in Supplementary Table 2. Sterilised *Arabidopsis* seeds were grown on plates containing 1/2 Murashige and Skoog (MS) medium containing 1% sucrose and supplemented with 1% Microagar. Seeds were stratified in the dark at 4 °C for 48 h. Plates were then transferred to climate chambers with a long day photoperiod (16 h light/8 h dark cycle) at 22 °C/18 °C and grown for 2 weeks. Light intensity during control growth conditions was ~100 µE m$^{-2}$ s$^{-1}$. The Columbia accession Col-0 was used as wild-type background for all experiments. For cold treatment, seedlings were grown in control conditions for two weeks and subsequently transferred to a cold room (4 °C) with light for 3 h before harvesting.

**TIF-seq library construction.** Discrete TU boundaries in *A. thaliana* were mapped by an optimised version of TIF-seq[44]. The TIF-seq approach was modified in order to better detect corresponding TSS/PAS pairs in organism more complex than that of budding yeast *Saccharomyces cerevisiae*. Most notably, the make-up and orientation of sequencing adapters was improved to effectively double the return of map-able reads. The detailed protocol will be published elsewhere[70]. Here we describe its application to *A. thaliana*. Briefly, total *Arabidopsis* RNA was isolated using the RNeasy Plant Mini Kit (Qiagen) and treated with Turbo DNase (ThermoFisher Scientific) according to manufacturers' instructions. DNase-treated RNA was recovered by acid–phenol extraction and ethanol precipitation. RNA integrity was assessed using the 2100 Bioanalyzer RNA 6000 Nano assay (Agilent). 10 micrograms of DNase-treated total RNA was treated with CIP (NEB) according to manufacturer's instructions in order to remove all non-capped RNA species in the sample. RNA was recovered by acid–phenol extraction and ethanol precipitation. Next, 5′ caps were removed using Cap-Clip (CellScript). RNA was quickly recovered by acid–phenol extraction and ethanol precipitation. The single-stranded rP5_RND adapter (see Supplementary Data 2 for oligonucleotide sequence) was ligated to the 5′-end of previously capped species using T4 RNA ligase 1 (NEB). RNA was recovered by AMPure purification using Agencourt RNAClean XP beads

(Beckman Coulter) following manufacturer's instructions. Before proceeding, RNA integrity was assessed using the 2100 Bioanalyzer RNA 6000 Nano assay (Agilent) following manufacturer's instructions. In order to generate full-length cDNAs up to and >10 kb long, SuperScript IV (Invitrogen) and an oligo(dT)-containing barcoded adapter (see Supplementary Data 2 for TIF2-RTX oligonucleotide sequences) was used with the following PCR steps: 10 min at 42 °C, 30 min at 50 °C, 30 min at 55 °C and 10 min at 80 °C. cDNA samples were treated with RNase H, provided with SuperScript IV First-Strand Synthesis System (Invitrogen), and recovered using AMPure XP beads (Beckman Coulter). Second-strand synthesis of full-length cDNA was amplified with Terra PCR Direct Polymerase Mix (Takara) the BioNotI-P5-PET oligo (see Supplementary Data 2 for oligonucleotide sequence) and the following PCR steps: 2 min at 98 °C, 10 cycles (20 s at 98 °C, 15 s at 60 °C, 5 min at 68°C + 10 s/cycle), 5 min at 72 °C. Barcoded full-length double-stranded cDNA libraries were recovered using AMPure XP beads (Beckman Coulter) and assessed for quality using the 2100 Bioanalyzer High Sensitivity DNA kit (Agilent). cDNA libraries were quantified with the Qubit dsDNA HS Assay Kit (ThermoFisher Scientific) according to manufacturer's instructions. Equal quantities of cDNA libraries (up to 600 ng total) were pooled and digested with NotI (NEB) for 30 min at 37 °C. The enzyme was inactivated at 65 °C. Digested cDNA was recovered using AMPure XP beads (Beckman Coulter). Digested linear cDNA pools were diluted to <1 ng/μl and circularised using T4 DNA ligase (NEB) at 16 °C overnight. PlasmidSafe™ DNase (Cambio) was added for 30 min at 37 °C to remove linear DNA molecules from the sample and then inactivated at 70 °C for 30 min. Circular cDNA molecules were recovered by phenol–chloroform extraction followed by isopropanol precipitation. Circular cDNA was sheared by sonication at 4 °C in a Bioruptor (Diagenode) (3 cycles, 30 s ON, 90 s OFF). Sheared DNA fragments were purified with AMPure XP beads (Beckmann Coulter). Biotinylated fragments containing flanking TSS/PAS pairs were captured with M-280 Streptavidin Dynabeads (ThermoFisher Scientific), end repaired with End Repair Enzyme mix (NEB), A-tailed with Klenow fragment exo- (NEB), and ligated to a common Illumina-compatible forked adapter using T4 DNA ligase (NEB) according to manufacturers' instructions. Libraries were amplified using the Phusion High-Fidelity PCR mix (NEB) with the following PCR steps: 98 °C for 1 min, 15 cycles (20 s at 98 °C, 30 s at 65 °C, 30 s at 72 °C), 5 min at 72 °C. Libraries were size selected using AMPure XP beads (Beckman Coulter) and sequenced with the following flowcell: FC-404-2002 NextSeq 500/550 High Output v2 kit (150 cycles) (Illumina). The sequencing run was carried out using a mix 1:1 of custom oligos SeqR1 + 15T and SeqR2 for sequence reads 1 and 2, and of custom oligos SeqINDX and TIF3-RvRT for the index reads 1 and 2 (see Supplementary Data 2 for oligonucleotide sequences). Cluster orientation was determined based on the information contained in the index reads. All TIF-seq experiments were performed as independent duplicates.

**TSS-seq library construction**. TSSs were mapped genome-wide in *Arabidopsis* using 5′-CAP-sequencing[71], with minor changes compared to detailed presentations of the method[8,35]. Note, we define this adapted method Transcription Start Site sequencing (TSS-seq). Here, we extend our previous TSS-seq analyses to 2-week-old *hen2-2* seedlings before and after cold treatment (3 h at 4 °C) in order to correct for TIF-TSS bias (see Computational methods below). All TSS-seq experiments were performed as independent duplicates.

**RT-qPCR**. RNA was isolated from two-week old *Arabidopsis* seedlings using Plant RNeasy Mini-Kits (Qiagen) and treated with Turbo DNase-treated (Ambion) as per manufacturers' instructions. For selective quantitative analysis of mRNAs and sppRNAs, first-strand complementary DNA (cDNA) synthesis was performed on RNA using an oligo(dT)-containing primer (see Supplementary Data 2 for oligonucleotide sequence) and SuperScript IV First-Strand Synthesis System (Invitrogen) following manufacturer's instructions. cDNA samples were treated with RNase H (Invitrogen) for 20 min at 37 °C before enzyme inactivation at 80 °C for 20 min. Negative controls lacking the reverse transcriptase enzyme (-RT) were performed alongside all RT-qPCR experiments. Quantitative analysis of mRNAs (F1/R1 primer pairs) or sppRNAs (F1/R2 primer pairs) were performed by qPCR using the CFX384 Touch Real-Time PCR Detection System (Biorad) and GoTaq qPCR Master Mix (Promega). See Supplementary Data 2 for oligonucleotide sequences used. Short annealing and extension times were used to specifically amplify and distinguish sppRNAs from their cognate mRNAs. qPCR cycles were as follows: 5 min at 98 °C, 40 cycles (10 s at 98 °C, 10 s at 55 °C, 5 s at 65 °C). Data were normalised to the mRNA levels of an internal reference gene lacking detectable sppRNAs (ACT2). Relative expression to wild type of mRNAs and sppRNAs were calculated with normalisation to ACT2 mRNA. Error bars reflect SEM resulting from at least three independent replicates. See source data file for values used to construct plots.

**Northern blotting**. Northern analysis in Fig. 3i was performed as previously described[35] with minor modifications. Briefly, 10 μg of total RNA was separated by electrophoresis on agarose–formaldehyde–MOPS gels and transferred to a nylon transfer membrane by capillary blotting in 6× SSC overnight. RNA was crosslinked to the nylon membrane by UV irradiation using a Stratalinker UV Crosslinker (Stratagene). Membranes were prehybridised in Church Buffer (0.5 M $Na_2HPO_4$

pH 7.2, 1% BSA, 1 mM EDTA, 7% SDS) for 1 h at 68 °C. Next, membranes were probed overnight at 68 °C with single-stranded DNA probes generated by PCR-based amplification and incorporation of radioactive α-$^{32}$P-dTTP (PerkinElmer). Membranes were washed twice in pre-warmed buffer containing 2× SSC and 0.1% SDS for 10 min (68 °C) before exposure to Storage Phosphor Screens (Kodak). A Typhoon phosphor-imager (GE Healthcare Life Sciences) was used for analysis. For loading controls, membranes were stripped three times with a near boiling buffer containing 0.1% SDS before prehybridization in Church Buffer as above and northern blotting against the mRNA of a reference gene lacking detectable sppRNAs (UBI). For northern analysis in Fig. 3j, 10 μg of total RNA and 1/1000 dilution of Decade™ Markers System ladder according to manufacturer instructions was separated by polyacrylamide gel electrophoresis (PAGE) with 6% urea, 0.5× TBE. The gel was blotted on an Amersham Hybond-NX nylon membrane (GE Healthcare Life Sciences) and crosslinked by UV irradiation. The membrane was probed as described above. For loading control, the reference snRNA U6 was end labelled with α-32P-ATP and hybridised at 42 °C. See Supplementary Data 2 for oligonucleotide sequences used to prepare radioactive DNA probe templates. Both experiments' northern blotting raw images can be found in the Source Data file.

**Nicotiana benthamiana leaf injection**. Transient translations of eYFP reporter constructs were generated based on pGWB540 vectors[72]. The promoter region (~1 kb) and the first exon until the ATG start codon of *MPK20* (pMPK20) and *ACT8* (pACT8) were amplified from *Arabidopsis* wild-type genomic DNA by PCR. The PCR products were ligated to the NotI-HF and AscI digested pENTR-D-Topo vector through isothermal assembly to yield entry clones. Short promoter-proximal RNA (sppRNAs) deletion constructs were generated from entry clones. The 58 bp sppRNA region in the first exon of the *MPK20* 5′-UTR was deleted (pMPK20-ΔsppRNA), and the 64 bp sppRNA region was deleted in the *ACT8* 5′-UTRs (pACT8-ΔsppRNA). Entry vectors were used in Invitrogen, Gateway LR Clonase II reaction according to manufacturer conditions with pGWB540 (enhanced YFP) to generate expression vectors: pMPK20-eYFP, pΔMPK20-eYFP, pACT8-eYFP and pΔACT8-eYFP. The expression vectors were transformed into *Agrobacterium tumefaciens* strain GV3850 by electroporation under 2.5 kV, 400 Ω and 25 μF. Agrobacteria containing expression vectors were respectively co-infiltrated with the p19 suppressor of silencing into *Nicotiana benthamiana* leaves. The eYFP signal was assayed on the third day after infiltration.

**Bioinformatics**. Computational analysis of TSS-seq data analysis was performed as previously described[35]. For TIF-seq, paired-end sequencing reads were demultiplexed and concatenated according to the indexes adjacent to 5′- or 3′-ends of transcript molecules. Reads were then subjected to adapter- and UMI-trimming with respectively cutadapt v1.18 ("AGGTGACCGG", "AGATCGGAAG") and UMI-Tools v0.5.5. Extra stretches of As near the polyadenylation sites were removed from reads using cutadapt v1.18. Next, reads containing 10-nt-long stretches of As and Ts with one mismatch were removed using AfterQC v0.9.7. The adapter-, UMI- and polyX-trimmed reads were aligned to TAIR10 genome assembly using STAR v2.6.1c (--alignEndsType EndToEnd --alignMatesGapMax 15000). The output sorted BAM files were filtered for uniquely aligned reads using SAMtools v1.7. Finally, the filtered BAM files were deduplicated using UMI-Tools dedup.py. The two replicate alignments of each data set were compared using deepTools multiBamsummary, calculating a coverage matrix between replicates (--binSize 10). The Pearson correlation for each matrix was plotted with the deepTools plotCorrelation (--corMethod pearson --log1p and --removeOutliers) in Supplementary Figs. 3c, 4a, b, 5a and 8a, b, showing reproducibility between replicates. Based on the reproducibility, the two replicates of TIF-seq were merged before clustering.

Each pair of reads aligned to the genome was clustered with the custom script "TIF-Seq_isoform_clustering_v6.py" as follows: "Pre-clusters" were called and read pairs overlapping exactly at both ends were merged. Next, Pre-clusters were merged if both start and end positions overlapped another existing "Pre-cluster" by <20 nucleotides. Subsisting merged pre-clusters were kept as final clusters only if their pairs of reads count was equal or higher than 3. The remainder were discarded. To further remove clusters produced by accidental 3′-end mispriming, clusters with extended end positions overlapping genomic polyA regions (from 6 to 9 As) called with bowtie aligner were discarded. Due to artefactual bias of TIF-seq called TSSs towards genic 3′-ends, as previously described[44], the 5′ reads of TIF-seq were used to call TIF-TSSs for each data set using the CAGEfightR package v1.0.0 package available from Bioconductor [https://bioconductor.org/packages/release/bioc/html/CAGEfightR.html] and filtered out with no overlap to a previously defined TSS called from published TSS-seq data[35]. The TSS peak distance to the annotated TSSs called with CAGEfightR for TSS-seq and TIF-seq in wild type, representing the TIF-TSSs bias towards small clusters overlapping the end of genes, later removed, is shown in Supplementary Fig. 3a, b. The overall numbers of paired reads along with discarded and final TIF-clusters called per genotype can be found in Supplementary Table 1.

For all further genome-wide clusters analysis, measurements considered only non-overlapping genes to avoid false-positive calculations while accepting to lose some information. In Figs. 1c, 3c and 4c, d, we calculated the exact distance between each cluster boundary (from the final TIF-clusters) to their corresponding overlapping gene boundaries extracted from TAIR10 coding genes annotation

available through the TxDb.Athaliana.BioMart.plantsmart28 R package (cf script "histscatter_all.R"). In the case of Fig. 4e, the same calculation was performed in *hen2-2* but only on clusters with widths <250 nucleotides (cf script "histscatter_small.R"). To analyse FACT TIF-clusters, we combined the two FACT mutants *ssrp1-2* and *spt16-1* TIF-seq data sets and extracted the fact-specific TIF-Seq clusters overlapping published *fact*-specific and upregulated TSSs—based on TSS-seq[35]. In Fig. 2f, we plotted the genomic distance of each TSS/PAS pairs between the subseted fact-specific TIF-seq clusters and their corresponding gene annotation. (cf script "histscatter_FACT.R"). For all boxplots, the box bounds the iQR divided by the median, and whiskers extend to a maximum of 1.5× iQR beyond the box. Data beyond the end of the whiskers are outliers and plotted as points. To evaluate the genome-wide distribution of clusters sizes between samples, all clusters <6000 nucleotides in width were plotted as a violin plot between wild type and *hen2-2* and in cold-treated wild type and cold *hen2-2* in Figs. 3d and 4f. To establish the positional distribution of isoforms across genes, the clusters boundaries were overlapped with the corresponding genes boundaries and segregated into different classes (TIF-cluster categories) with specific comparison to annotated genes using the countOverlaps() function from R package "Genomic Ranges" (cf script "sepclusters_orf_simplified.R" and "sepclusters_overlapping_orf. R"). Each number for TIF-cluster categories were compared to the ensemble of all genes overlapping TU clusters. These analyses are represented as piecharts in Figs. 1d, 2a, b, 3a and 4a, b. The log2 fold-change of each category across genotype in wild type and *hen2-2* was calculated in Fig. 3b and between cold-treated wild type and *hen2-2* in Supplementary Fig. 8e. The number of clusters overlapping the whole ORF was measured for each gene in each data set and plotted as histograms in Supplementary Figs. 3d, 4b–d and 8c, d.

To extract small promoter-proximal clusters, all clusters <350 nucleotides were mapped to annotated TSSs extended by 40 nucleotides upstream and downstream and to overlap no termination sites to account for TSS variation. The corresponding set of overlapping clusters were then called short promoter-proximal RNAs (sppRNAs; see Supplementary Data 3 for full list of genomic coordinates). sppRNA size distribution was plotted in Fig. 3e (cf script "sepclusters_forsmallclusters.R").

All further analysis requiring expression controls were calculated using transcription profiles of published unphosphorylated RNAPII (8WG16) pNET-seq wild-type data sets[43]. pNET-seq signal was averaged over all genes shrunk by 200 nucleotides downstream of the TSS (+200 from TSS) and 200 nucleotides upstream of the PAS (−200 from TSS) to reduce biases from recently reported peaks in nascent transcription signals near TSS-proximal or PAS-proximal positions[43]. The percentage of cold-induced genes (calculated from wild-type TSS-seq data) that only have sppRNAs at 22 °C was compared against a control set of genes without detectable sppRNAs in Supplementary Fig. 9a, b; all sppRNA-containing genes common between wild type and *hen2-2* before and after cold treatment were plotted as a Venn Diagram (cf script "ViennDiagram.R").

The distribution of nascent transcriptional activity over sppRNA genes was compared to all other genes with no sppRNAs in wild type in Fig. 3f. A set of genes with no sppRNAs detected in any data set and with equal distribution of nascent gene body transcription (pNET-seq) was calculated using the script "subset_equaldistribution_v4.R" and used as control set of genes. Signal of pNET-seq was plotted for both the sppRNA genes and control data sets in Fig. 5c anchored at the TAIR10 annotated TSSs. The nascent transcription signal downstream of the TSS was averaged for the two genes sets and their distribution compared by Wilcoxon test. Main genes TSS positions from TSS-seq were used to identify *cis*-elements in promoter DNA sequences and gene body sequences. Promoter DNA sequences were defined as 300 nt upstream and 50 nt downstream of the measured TSS, and downstream sequences were defined as 20 nt upstream of the gene TSS and 250 nt downstream. Enriched *cis*-elements between the two control sets were identified with MEME suite DREME and RSAT web tools by differential comparison of sequences regions for sppRNA genes and control genes of similar nascent transcription distribution enrichment and *p*-values in source data file. The main TSSs were used as anchor points in the metagene plots in Fig. 5a, c and *cis*-elements heatmaps in Fig. 5d, f and Supplementary Fig. 13a–c TSSs were described in ref. [35]. In Fig. 5a, the termination sites of sppRNAs (i.e. the PAS) were used as anchor for the metagene plot. The number of sppRNAs genes per 100 genes in relation to the 5′ pausing index value in Supplementary Fig. 12 was plotted for 12,854 non-overlapping genes longer than 1000 nt and belonging to the 75% most expressed genes. The 5′ pausing index value was calculated as previously described[43] by dividing the promoter TPM coverage from pNET-seq (150 nt upstream and downstream around the TSS) to the gene body (shrunk gene region by 300 nt downstream of the TSS and 300 nt upstream of the PAS). Full-length mRNA expression was compared between mutants by measuring TPM at genes annotated PAS site from PAT-seq experiments. A specific control set of genes with equal distribution to sppRNA genes was calculated with the corresponding experiment wild-type samples, then full-length expressions in TPM were compared by Wilcoxon test between the control set of genes and sppRNA genes in the mutant sample and in wild type in Fig. 6c, d.

Finally, all genome browser screenshots were made using Integrative Genomics Viewer (IGV_2.3.93) with the *A. thaliana* (TAIR10) annotation. In particular, TIF-seq reads (BAM alignment files, before clustering) were visualised with the following settings: collapsed, view as pairs, colour alignments by read strand.

**Reporting summary**. Further information on research design is available in the Nature Research Reporting Summary linked to this article.

## Data availability
The data that support this study is available from the corresponding author upon reasonable request. TIF- and TSS-seq data is available at NCBI GEO database with accession code GSE129523. Previously published TSS-seq data is available from GEO accession number GSE113677. The source data underlying Figs. 1d, e, 2a, b, 3a, b, h–j, 4a, b, 6a and Supplementary Figs. 6, 7, 8e, 9b, c, 10e, 13d, f, and 15a, b are provided as a Source Data file, whereas genomic coordinates for sppRNA TIF-clusters per genotype and environmental conditions are listed in Supplementary Data 3.

## Code availability
TIF- and TSS-seq analysis computational methods scripts and files scripts can be freely accessed at github [https://github.com/qthom/planTIF]. Bed files for each TIF-seq data set representing read pairs mapped before and after clustering are available to be used in scripts or can be uploaded into a genome browser.

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

## Acknowledgements

We thank members of the Marquardt and Pelechano laboratories for discussions and technical assistance. We thank members of the P. Brodersen lab for help with the PAGE northern. We are grateful to T.H. Jensen, S. Buratowski and P. Brodersen for critically evaluating the manuscript. Research in the Marquardt lab is supported the Novo Nordisk Foundation (NNF15OC0014202), and a Copenhagen Plant Science Centre Young Investigator Starting Grant. This project has received funding from the European Research Council and the Marie Curie Actions under the European Union's Horizon 2020 research and innovation programme (StG2017-757411) (S.M.). R.A. was supported by an European Molecular Biology Organization Long-Term Fellowship (ALTF 463-2016). The Pelechano lab is supported by a SciLifeLab Fellowship (Karolinska Institutet SFO-PRIO), the Swedish Research Council (VR 2016-01842), a Wallenberg Academy Fellowship (KAW 2016.0123) and the Ragnar Söderberg Foundation.

## Author contributions

S.M. and V.P. conceived the project with input from all authors; S.M., V.P., Q.T. and R.A. designed experiments. Q.T, R.A. and L.J. performed the experiments. Q.T. performed computational analyses with support from J.W. and V.P. B.L., J.W. and V.P. optimised TIF-seq. S.M. supervised the project. S.M. wrote the manuscript with input from all authors.

## Competing interests

The authors declare no competing interests.
