## [Peer Review File · Nature Communications]

Reviewers' comments:

Reviewer #1 (Remarks to the Author):

In this manuscript, the authors use a new sequencing technique to simultaneously determine the transcription start sites (TSSs) and polyadenylation sites (PASs) of transcripts: Transcript Isoform sequencing (TIF-seq). Current techniques exist to identify the 5' or 3' ends present in a population of transcripts, but it remains difficult to determine exactly which 5' and 3' end site exist in an individual molecule. The authors use TIF-seq to investigate transcription in Arabidopsis, using both wild type and backgrounds with defective RNA degradation. The later allows for the detection of short-lived transcripts that would be otherwise quickly degraded. This allowed for the discovery of ~4 isoforms per expressed gene with ~14% of gene expressing unstable short promoter-proximal RNAs (sppRNAs). Mutations in elongation factors increase the ratio of sppRNAs to full-length mRNAs, suggesting that Pol II stalling may contribute to the production of sppRNAs.

The results presented here make important contributions to our understanding of transcriptional control. In particular, the presence of sppRNAs draws attention to the importance of elongation factors in assisting Pol II in the early phases of transcriptional elongation. The main weakness of the manuscript is that the data is fairly descriptive. It is not clear how sppRNAs might contribute to gene regulation in the normal life cycle or whether sppRNAs are present in other plants or animals. Additional experiments in either of these veins would increase the significance and impact of the work.

Regarding the writing, the manuscript would be improved by taking advantage of the longer page limits afforded by this journal. A clear introduction and a longer discussion of the significance of the results would be very helpful.

Reviewer #2 (Remarks to the Author):

The manuscript by Ard et al performs TIF-seq (transcript isoform sequencing, a high throughput sequencing technique that captures both 5' and 3' end of polyadenylated RNA molecules) in whole Arabidopsis seedlings, comparing wild type to knockout effects of several genes involved in transcriptional elongation, RNAPII stalling and pre-mRNA cleavage. The main finding is that there is a genome-wide occurrence of unstable short promoter-proximal RNAs (sppRNA), and that the knockouts of different genes involved in the aforementioned functions result in either the increase or decrease of their level relative to the mRNA expression.

The main findings are of interest to the broader community doing research on promoter function and transcriptional elongation. There are number of issues, though:

- The paper is written in a condensed letter format, which suggests that it was originally meant for a different journal with much more severe space restrictions than Nature Communications. In this case, brevity is not helping. The literature review of current knowledge as well as discussion of the implications of the results are rudimentary, and some of the results central to the flow of the paper refer exclusively to supplementary figures. I suggest to convert it into a full length paper to make the readers' orientation easier, and to provide a proper context for the reported results. (Some concrete suggestions follow.)

- The paper needs a proper introduction into what is already known about transcript heterogeneity at 5' and 3' ends, both in plants and the parallels with Metazoan genomes. The heterogeneity at the 5' ends is practically universal in Metazoan genomes, and multiple polyadelylation signals are common. As the authors remark in the passing, early polyadenylation signals play a central role in early termination of antisense transcripts in promoter architectures

with bidirectional initiation but a functional transcript in only one of the two directions.

- Further, it is well known that promoter-proximal pausing and dispersed transcription initiation positions within a promoter are related to specific (not all) promoter architectures (typically TATA-less, broadly expressed and some developmentally regulated promoters). It is a missed opportunity, and not a difficult one to explore, to investigate which promoter elements might correlate with more or less sppRNA production as well as the number of isoforms. If TIF-seq does not have enough coverage for a good single-nucleotide resolution at 5' ends of genes, the TSS-seq could be used to determine dominant TSS positions more precisely.

- Since many sppRNAs are associated with highly expressed genes, at least some of which are ribosomal protein genes and other components of transcriptional machinery, it would be especially interesting to see if they have a separate promoter architecture like they do in Metazoa, including the TCT initiator (for details see e.g. review by Kadonaga, WIREs Dev Biol 2012).

- Do multiple isoforms result in changes in first splice site of the transcript? Do sppRNAs prefer transcripts with longer or shorter first exons?

Minor:

- Nonsense-mediated decay is a mechanism by which many aberrant transcripts are removed in metazoan transcriptomes. Any hints of its role in sppRNA degradation?

Reviewer #3 (Remarks to the Author):

In this manuscript Ard and colleagues analyse the genomic distribution and abundance of RNA polymerase II (Pol II) initiation and termination events in *Arabidopsis thaliana*. They report evidence for unstable short promoter-proximal RNAs (sppRNAs) at ~14% of expressed genes, and an average of four different transcript isoforms per gene in wild-type (WT) plants. These results were enabled by an improved Transcript Isoform Sequencing (TIF-seq) protocol based on a coauthor's published method (Pelechano et al. Nature 2013; Pelechano et al. Nat Protoc. 2014), which sequences cDNA tags from matching transcription start sites (TSSs) and polyadenylation sites (PASs) of Pol II transcripts. The authors' global analysis of *Arabidopsis* transcript TSS/PAS pairs, using mutants known to be defective in transcriptional regulation or transcript degradation, offers some new insights that will be of interest to molecular biologists in this field. However, the study has weaknesses that should be remedied prior to consideration for publication.

Performing TIF-seq on the *hen2-2* mutant, which is defective for nuclear exosome activity (Lange et al. 2014 PLoS Genet), Ard and colleagues detect evidence for transcripts initiating at the annotated TSS but terminating <100 nt downstream, on average. These sppRNAs were putatively confirmed at the *MPK20* gene when a <500 nt smear was detected via northern blot using a single probe (Fig. 2(e)). This seems to substantiate TIF-seq data shown in Fig. 2(d), but to confirm the size-range and gene position of such sppRNAs further experiments are needed (Major concern #1).

Much of the remaining work in this study is of high technical quality, but the manuscript's clarity suffers from the relegation of clear examples to supplemental figures, with more obscure data displays shown as primary figures. For instance, the *AT5G51200* (Fig. S4(g)) and *AT4G15260* (Fig. S4(h)) loci illustrate the authors' point that the FACT complex represses alternative TSSs, whereas the scatterplot of primary Fig. 1(f) requires careful study and a detailed reading of the Methods to interpret (Major concern #2).

Finally, the authors' Page 5 statement that, "most genes with sppRNA show no evidence for gene regulation by selective termination and an equivalent fraction of mRNA without sppRNA are cold-induced" is an accurate summary of the authors' data: the function of such sppRNAs in gene regulation, if any, remains quite enigmatic. The authors should avoid masking this point with speculative conclusions (Major concern #3).

Major concerns:

1) Page 4 and Fig. 2(e): Indistinct smears are frequently detected in northern blots due to unequal loading or other artifacts of RNA preparation. Furthermore, the small expected size of sppRNAs (median 93 nt) means that >50% of the RNAs are shorter than can be resolved via formaldehyde-agarose electrophoresis (the technique used here). I suggest that authors reproduce their northern result using polyacrylamide gel electrophoresis (PAGE) with appropriate RNA size standards in order to confirm the size range of sppRNAs. For both the formaldehyde-agarose and PAGE northern blots an additional probe could be hybridized to detect MPK20 mRNAs via a 3' region not overlapping sppRNAs. If the authors' hypothesis is correct, then this second probe should detect full-length MPK20 isoforms in WT and hen2-2 samples but not the putative sppRNAs in hen2-2.

2) Page 3 and Fig. 1: I recommend that supplemental panels Fig. S4(g) and Fig. S4(h) be included in primary Fig. 1, because these clearly illustrate FACT suppression of intragenic Pol II initiation. Conversely, the Fig. 1f scatterplot should be revised because the underlying data and analyses are unclear: precisely how were the two fact mutants (spt16-1 and ssrp1-2) analysed? Did these two mutants differ? Were replicate experiments conducted? How were WT/mutant comparisons handled? How was the threshold for inclusion in the scatterplot chosen? Were any statistical analyses performed? These info should be in the results and legend (not buried in Methods), because they are essential for readers to interpret the figure.

3) Page 5: The authors do not present data supporting the speculative statement that concludes this paragraph: "...promoter-proximal termination is associated with plant gene expression across temperature and may contribute to temperature-dependent gene regulation." The first half of the sentence refers to sppRNAs being co-expressed with mRNAs at cold-induced genes (a simple correlation), but the second half contradicts the overall TIF-seq analysis as presented and summarized by the authors in this same paragraph.

Minor points/corrections

1) Abstract: "... how alternative TSSs connect to variable PASs is unresolved from common transcriptomics methods," would better read, "... how alternative TSSs connect to variable PASs is not resolved by common transcriptomics methods."

2) Page 2: To illustrate functionally distinct mRNA isoforms, I suggest that the authors cite the N-terminal nuclear localisation signal of Dicer-like 4 (DCL4) in *A. thaliana*. Alternative TSS selection that depends on promoter DNA methylation allows different isoforms to be expressed from the single DCL4 gene (Pumplin et al. 2016 Plant Cell).

3) Page 3: There is a definite article missing here: "The detection of many RNA species that are produced in wild type yet rapidly degraded...", should read, "The detection of many RNA species that are produced in the wild type yet rapidly degraded...".

The text of the comments we received by the reviewers is marked in **blue**, our response is marked in **black**.

Reviewer #1 (Remarks to the Author):

In this manuscript, the authors use a new sequencing technique to simultaneously determine the transcription start sites (TSSs) and polyadenylation sites (PASs) of transcripts: Transcript Isoform sequencing (TIF-seq). Current techniques exist to identify the 5' or 3' ends present in a population of transcripts, but it remains difficult to determine exactly which 5' and 3' end site exist in an individual molecule. The authors use TIF-seq to investigate transcription in *Arabidopsis*, using both wild type and backgrounds with defective RNA degradation. The later allows for the detection of short-lived transcripts that would be otherwise quickly degraded. This allowed for the discovery of ~4 isoforms per expressed gene with ~14% of gene expressing unstable short promoter-proximal RNAs (sppRNAs). Mutations in elongation factors increase the ratio of sppRNAs to full-length mRNAs, suggesting that Pol II stalling may contribute to the production of sppRNAs.

The results presented here make important contributions to our understanding of transcriptional control. In particular, the presence of sppRNAs draws attention to the importance of elongation factors in assisting Pol II in the early phases of transcriptional elongation. The main weakness of the manuscript is that the data is fairly descriptive. It is not clear how sppRNAs might contribute to gene regulation in the normal life cycle or whether sppRNAs are present in other plants or animals. Additional experiments in either of these veins would increase the significance and impact of the work.

We thank reviewer 1 for appreciating the importance of our findings. We were able to strengthen our manuscript further by adding experiments addressing both veins.

1.) To address whether sppRNA equivalents are present in other system we highlight the similarities and differences of sppRNA in other systems more prominently in our revised manuscript. Fortuitously, pre-RNA processing of promoter-proximal RNA species that bear some resemblance to sppRNA were described in *Drosophila* while our manuscript was under review (PMIDs: 31809743, 31530651). We added the relevant citations and highlighted these novel parallels to metazoans, for example in the discussion.

- Since the Integrator complex is linked to promoter-proximal transcriptional termination in *Drosophila* (PMIDs: 31809743, 31530651), we are also able to add experimental support for these similarities. We have added an analysis of sppRNA in two *Arabidopsis* Integrator mutants in revised Figure 6. Our data support a role for integrator in promoter-proximal termination of RNAPII transcription, adding experimental support for conserved elements mediating promoter-proximal termination to strengthen the manuscript.

- We strengthened our revised manuscript by providing genome-wide support for a contribution of CPSF/CstF in mRNA expression of sppRNA genes. In our initial submission we found that *cstf64-2* mutants impaired the ratio of sppRNA/mRNA termination, resulting in increased full length mRNA by RT-qPCR. To test this hypothesis

genome-wide, we re-analyzed published PAT-seq data of two CPSF/CstF mutants, known mediators of mRNA polyadenylation (*CstF77* and *CPSF100*). We compared the expression of canonical poly-(A) sites between sppRNA genes and control genes without sppRNAs. We observed a specific genome wide increase of full length mRNA for sppRNAs genes. This data is represented in figure 6 c-d. These data support a role for sppRNA termination on full length mRNA expression, akin to the “attenuation” mechanism suggested for metazoans. Future research will be necessary to fully resolve the contributions of Integrator and CPSF/CstF in sppRNA formation.

- Our analyses of cis-elements suggested by reviewer #2 uncovered an additional similarity to metazoan promoter-proximal transcriptional termination. We uncovered the GAGA-motif in novel computational analysis that is added as revised Figure 5d-g and Supplementary figure 13. The GAGA-motif is linked to promoter-proximal RNAPII stalling in drosophila. We have expanded on these similarities indicating conservation of sppRNA in the revised manuscript text.

2.) To address gene regulation by sppRNA through additional functional data, we assayed the effect of mutating sppRNA in the 5'-UTRs of genes and assayed the effect of reporter gene expression. We added these data as novel figure, revised supplementary figure 10. The data suggest that sppRNA may promote gene expression, consistent with the genome-wide positive correlation between sppRNA detection and gene expression. While further research will be needed to fully resolve the roles of sppRNA in gene regulation in more detail, we hope reviewer #1 can appreciate that these data strengthen our manuscript.

Regarding the writing, the manuscript would be improved by taking advantage of the longer page limits afforded by this journal. A clear introduction and a longer discussion of the significance of the results would be very helpful.

We address this comment with substantial revisions to the text, structure and layout. We believe they capture the essence of this comment and strengthen the manuscript.

Reviewer #2 (Remarks to the Author):

The manuscript by Ard et al performs TIF-seq (transcript isoform sequencing, a high throughput sequencing technique that captures both 5' and 3' end of polyadenylated RNA molecules) in whole Arabidopsis seedlings, comparing wild type to knockout effects of several genes involved in transcriptional elongation, RNAPII stalling and pre-mRNA cleavage. The main finding is that there is a genome-wide occurrence of unstable short promoter-proximal RNAs (sppRNA), and that the knockouts of different genes involved in the aforementioned functions result in either the increase or decrease of their level relative to the mRNA expression.

The main findings are of interest to the broader community doing research on promoter function and transcriptional elongation. There are number of issues, though:

- The paper is written in a condensed letter format, which suggests that it was originally meant for a different journal with much more severe space restrictions than Nature Communications. In this case, brevity is not helping. The literature review of current knowledge as well as discussion of the implications of the results are rudimentary, and some of the results central to the flow of the paper refer exclusively to supplementary figures. I suggest to convert it into a full length paper to make the readers' orientation easier, and to provide a proper context for the reported results. (Some concrete suggestions follow.)

We apologize for the inappropriate manuscript format. We fully followed the suggestions of reviewer #2. We agree that the recommended revisions to our manuscript will make it more accessible to a broad audience.

- The paper needs a proper introduction into what is already known about transcript heterogeneity at 5' and 3' ends, both in plants and the parallels with Metazoan genomes. The heterogeneity at the 5' ends is practically universal in Metazoan genomes, and multiple polyadenylation signals are common. As the authors remark in the passing, early polyadenylation signals play a central role in early termination of antisense transcripts in promoter architectures with bidirectional initiation but a functional transcript in only one of the two directions.

We have expanded the introduction substantially according to the suggestions by reviewer #2. The important parallels to metazoan transcriptional regulation are indeed very informative and are now more clearly accessible in the revised manuscript.

- Further, it is well known that promoter-proximal pausing and dispersed transcription initiation positions within a promoter are related to specific (not all) promoter architectures (typically TATA-less, broadly expressed and some developmentally regulated promoters). It is a missed opportunity, and not a difficult one to explore, to investigate which promoter elements might correlate with more or less sppRNA production as well as the number of isoforms. If TIF-seq does not have enough coverage for a good single-nucleotide resolution at 5' ends of genes, the TSS-seq could be used to determine dominant TSS positions more precisely.

We thank reviewer 2 for this excellent suggestion. The suggested analyses are now included in Figure 5d-g and supplementary figure 13A-F in the revised manuscript. Perhaps surprisingly, we could not uncover differences in the TATA signature. However, we identified that the TCP transcription factor binding motif is enriched upstream of sppRNA genes. Moreover, we find an enrichment of the GAGA-box. Since the GAGA-box is linked to promoter-proximal pausing in metazoans, these analyses represented a nice opportunity to further strengthen the connections to the metazoan literature. Interestingly, the positioning of the GAGA-box is different in metazoans, it is shifted to positions largely downstream of the TSS in plants. The new computational analyses offer new insight and strengthen the revised manuscript.

- Since many sppRNAs are associated with highly expressed genes, at least some of which are

ribosomal protein genes and other components of transcriptional machinery, it would be especially interesting to see if they have a separate promoter architecture like they do in Metazoa, including the TCT initiator (for details see e.g. review by Kadonaga, WIREs Dev Biol 2012).

Our analyses could not identify a specific motif such as the TCT initiator. The computational analyses included in our revised manuscript suggest that promoters of genes with sppRNA are enriched for the TCP motif upstream of the TSS, and the GAGA-box largely downstream of the TSS. We hope the additional data clarify some of the questions concerning differences in promoter architecture.

- Do multiple isoforms result in changes in first splice site of the transcript? Do sppRNAs prefer transcripts with longer or shorter first exons?

Unfortunately, TIF-Seq is not well suited to resolve information on splice sites and we were unable to perform the suggested analysis regarding splice sites. To address the question about sppRNA termination, we tested for a biased location of sppRNA termination sites in introns or exons in Figure 1 below. We observed no clear bias in the termination site of sppRNA. sppRNA termination may occur in the 5'-UTR, 1st exon and 1st intron (left panel). When plotted, we observe a slightly shorter first exon in sppRNA genes Figure 1 (right panel). It would be interesting to follow-up on this observation in future studies. Currently, we feel that this information is best released in this document to satisfy the curiosity of reviewer #2.

Figure 1 : (Left) Proportion of termination sites for sppRNAs. sppRNA terminate in the 5'UTR, the coding exon of the first exon or in the first Intron. (right) First exon length distribution in sppRNA genes and a control set of genes with equal native expression. The length distributions were compared and statistical significance p-value calculated with the Wilcoxon test.

Minor:

- Nonsense-mediated decay is a mechanism by which many aberrant transcripts are removed in metazoan transcriptomes. Any hints of its role in sppRNA degradation?

To address this comment we requested and received seeds of *Arabidopsis* NMD mutants from the Riha lab (PMID: 22379136). *Arabidopsis* NMD mutants display auto-immunity phenotypes resulting in severe growth defects, confounding simple mutant vs wild type comparisons. The growth defects of NMD mutants are connected to auto-immunity and can be suppressed by blocking disease signaling through mutations in the *PAD4* gene. To control for growth defects of NMD mutants, we compared the effect of NMD mutants in the *pad4* mutant background (i.e. *smg7/pad4* against the *pad4* single mutant). We isolated *smg7-1/pad4-1* homozygous double mutants and *pad4-1* single mutants from a segregating population (as in PMID: 22379136). We extracted RNA from leaf and measured expression levels of mRNA and sppRNA for the target genes described in the manuscript by RT-qPCR. However, we fail to detect specific effects of this NMD mutant in relation with sppRNAs. We include these analyses below in Figure 2 for the information of reviewer #2.

Figure 2: (left) Relative expression of sppRNA normalized to actin for *smg7-1/pad4-1* and *pad4-1* for 4 genes with sppRNAs used in the manuscript (*HSC70*, *MPK20*, *RLP18e*, *STV1*). (right) Relative expression normalized to actin of full length mRNA for *smg7-1/pad4-1* and *pad4-1* for 4 genes with sppRNAs used in the manuscript (*HSC70*, *MPK20*, *RLP18e*, *STV1*).

Reviewer #3 (Remarks to the Author):

In this manuscript Ard and colleagues analyse the genomic distribution and abundance of RNA polymerase II (Pol II) initiation and termination events in *Arabidopsis thaliana*. They report evidence for unstable short promoter-proximal RNAs (sppRNAs) at ~14% of expressed genes, and an average of four different transcript isoforms per gene in wild-type (WT) plants. These results were enabled by an improved Transcript Isoform Sequencing (TIF-seq) protocol based on a coauthor's published method (Pelechano et al. Nature 2013; Pelechano et al. Nat Protoc. 2014), which sequences cDNA tags from matching transcription start sites (TSSs) and polyadenylation sites (PASs) of Pol II transcripts. The authors' global analysis of *Arabidopsis* transcript TSS/PAS pairs, using mutants known to be defective in transcriptional regulation or transcript degradation, offers some new insights that will be of interest to molecular biologists in this field. However, the study has weaknesses that should be remedied prior to consideration for publication.

To thank reviewer #3 for the appreciation of our new insights. We are grateful for the clear suggestions and outlined how we have used them to improve our manuscript below.

Performing TIF-seq on the *hen2-2* mutant, which is defective for nuclear exosome activity (Lange et al. 2014 PLoS Genet), Ard and colleagues detect evidence for transcripts initiating at the annotated TSS but terminating <100 nt downstream, on average. These sppRNAs were putatively confirmed at the *MPK20* gene when a <500 nt smear was detected via northern blot using a single probe (Fig. 2(e)). This seems to substantiate TIF-seq data shown in Fig. 2(d), but to confirm the size-range and gene position of such sppRNAs further experiments are needed (Major concern #1).

Much of the remaining work in this study is of high technical quality, but the manuscript's clarity suffers from the relegation of clear examples to supplemental figures, with more obscure data displays shown as primary figures. For instance, the *AT5G51200* (Fig. S4(g)) and *AT4G15260* (Fig. S4(h)) loci illustrate the authors' point that the FACT complex represses alternative TSSs, whereas the scatterplot of primary Fig. 1(f) requires careful study and a detailed reading of the Methods to interpret (Major concern #2).

Finally, the authors' Page 5 statement that, "most genes with sppRNA show no evidence for gene regulation by selective termination and an equivalent fraction of mRNA without sppRNA are cold-induced" is an accurate summary of the authors' data: the function of such sppRNAs in gene regulation, if any, remains quite enigmatic. The authors should avoid masking this point with speculative conclusions (Major concern #3).

Major concerns:

1) Page 4 and Fig. 2(e): Indistinct smears are frequently detected in northern blots due to unequal loading or other artifacts of RNA preparation. Furthermore, the small expected size of sppRNAs (median 93 nt) means that >50% of the RNAs are shorter than can be resolved via formaldehyde-agarose electrophoresis (the technique used here). I suggest that authors reproduce their northern result using polyacrylamide gel electrophoresis (PAGE) with appropriate RNA size standards in

order to confirm the size range of sppRNAs. For both the formaldehyde-agarose and PAGE northern blots an additional probe could be hybridized to detect MPK20 mRNAs via a 3' region not overlapping sppRNAs. If the authors' hypothesis is correct, then this second probe should detect full-length MPK20 isoforms in WT and *hen2-2* samples but not the putative sppRNAs in *hen2-2*.

We thank reviewer 1 for pointing out this deficiency of our analysis. We have expanded our characterization of sppRNA by northern blotting.

1.) We provide a new agarose northern blot experiment that is shown as revised Figure 5i. We improved sample loading and used the additional probe specific to the 3'-end. These data improve our manuscript since sppRNA detection is specific to the probe specific to the 5'-end, as reviewer #3 suggests.

2.) We include the requested PAGE northern as panel j of our revised Figure 5. We end-labeled a size marker to resolve the size distribution of sppRNA with improved resolution. As reviewer #3 suspected, the PAGE northern resolves sppRNA in a size range consistent with our estimate based on TIF-seq data. Our new experimental data clarify the size range and position of sppRNAs.

2) Page 3 and Fig. 1: I recommend that supplemental panels Fig. S4(g) and Fig. S4(h) be included in primary Fig. 1, because these clearly illustrate FACT suppression of intragenic Pol II initiation. Conversely, the Fig. 1f scatterplot should be revised because the underlying data and analyses are unclear:

- precisely how were the two fact mutants (*spt16-1* and *ssrp1-2*) analysed?
- Did these two mutants differ?
- Were replicate experiments conducted?
- How were WT/mutant comparisons handled?
- How was the threshold for inclusion in the scatterplot chosen?
- Were any statistical analyses performed?

These info should be in the results and legend (not buried in Methods), because they are essential for readers to interpret the figure.

- **We analyzed the FACT mutant TIF-Seq datasets the same as the other datasets in the manuscript. This is now more pointed out more clearly in our revised manuscript through an improved description and representation.**
- **We followed the excellent suggestion to include the TIF-seq data in FACT mutants as a main Figure. We add an improved representation of these data as revised Figure 2.**
- **Two biological repeats TIF-seq experiments were performed each. Two repeats of *spt16-1* mutants and two repeats of *ssrp1-2* mutants. The correlation between these datasets is shown in revised Supplementary Figure 4.**
- **We previously identified intragenic TSSs upregulated in both mutants of the FACT complex (i.e. *fact*-specific TSSs defined in PMID: 30707695). Differential expression analysis (with DESeq) was performed in TSS-Seq datasets between both *fact* genotype and wild type Col-0 seedlings separately. Then, intragenic up-regulated TSSs common**

to the two genotypes were defined as *fact*-specific intragenic TSSs. These analyses were performed with two biological replicate TSS-seq data for each genotype. (PMID: 30707695).

- In the case of the scatterplot in Figure 2f, the threshold to include TIF clusters in the scatterplot is a simple and exact overlap of the TIF-TSSs positions to the *fact*-specific TSSs positional windows. To increase the quality of our analyses, we restricted Figure 2f to intragenic TSSs detected in TIF-seq that we had previously identified with high confidence by TSS-seq. A condition to identify *fact*-specific TSSs was the identification of these intragenic TSSs in both mutants in the *FACT* complex. Each TSS (that we identified by TSS-seq) is associated with a genomic window by the CAGEfightR R library and identified based on high expression and dispersion in the whole dataset PMID: 31585526. To simplify the representation and to focus the analyses on the effect of the *FACT* complex we combined the two TIF-Seq datasets for the representation of Figure 2f. The panel in Figure 2f thus provides a condensed overview of the 3'-end positions of transcription events that initiate within transcription mutants when the function of the *FACT* complex is compromised.

We improved the legend for Figure 2f as requested, clarified the manuscript text and improved the method description.

3) Page 5: The authors do not present data supporting the speculative statement that concludes this paragraph: "...promoter-proximal termination is associated with plant gene expression across temperature and may contribute to temperature-dependent gene regulation." The first half of the sentence refers to sppRNAs being co-expressed with mRNAs at cold-induced genes (a simple correlation), but the second half contradicts the overall TIF-seq analysis as presented and summarized by the authors in this same paragraph.

We have revised the presentation of these results. Overall, sppRNA formation correlates with nascent transcription level as reviewer #3 points out. However, our analyses addressing potential regulation through selective sppRNA formation during revealed data consistent for some (38 of 1153) loci (line 250). To avoid misleading claims regarding gene regulation by sppRNA in our manuscript, we have revised the presentation of these data. We have included these numbers in the text so that readers will be in the position to judge this for themselves. It would clearly be interesting to explore the function of sppRNA in general and at specific loci in future studies.

Nevertheless, the PAT-seq data analyses in CstF/CPSF mutants provided in the revised Figure 6 are consistent with the possibility of mRNA regulation by "attenuation" as suggested in metazoans. The sppRNA deletion experiments provided as revised supplementary Figure 10 support the idea that sppRNA may participate in gene activation. We do not see it as a contradiction that sppRNA may regulate mRNA expression through "attenuation" and may participate in gene activation. We have elaborated on that point in our revised discussion. The purpose of sppRNA will have to be fully resolved in future studies but our revised manuscript offers tantalizing starting hypotheses for the field.

Minor points/corrections

1) Abstract: "... how alternative TSSs connect to variable PASs is unresolved from common transcriptomics methods," would better read, "... how alternative TSSs connect to variable PASs is not resolved by common transcriptomics methods."

We have revised the confusing sentence in the abstract and replaced it with a new sentence line 21.

2) Page 2: To illustrate functionally distinct mRNA isoforms, I suggest that the authors cite the N-terminal nuclear localisation signal of *Dicer-like 4 (DCL4)* in *A. thaliana*. Alternative TSS selection that depends on promoter DNA methylation allows different isoforms to be expressed from the single *DCL4* gene (Pumplin et al. 2016 Plant Cell).'

We have included a citation to this excellent publication. Unfortunately, DCL4 expression level in seedlings seems rather low, and our coverage of this locus by TIF-seq does not offer an improved screenshot for the revised manuscript. Since reviewer #3 may be interested in small RNA biogenesis more broadly, we add a TIF-seq data screenshot of AGO1 below. The AGO1 gene represents a sppRNA gene with a relatively high rate of sppRNAs.

3) Page 3: There is a definite article missing here: “The detection of many RNA species that are produced in wild type yet rapidly degraded...”, should read, “The detection of many RNA species that are produced in the wild type yet rapidly degraded...”.

Thank you for pointing the mistake, we have changed (line 64-65) to “Transcriptome analyses in nuclear exosome mutants facilitate the detection of many cryptic RNA species”.

REVIEWERS' COMMENTS:

Reviewer #2 (Remarks to the Author):

The authors have addressed most of my early concerns adequately. The new analysis of motifs is especially interesting. There are a couple of outstanding issues:

- The Introduction has been expanded as other reviewers and myself suggested, but it hasn't been done in the most careful way. For example, the sentence:

"The precise positions of TSSs may form a "focused" pattern with one predominant TSS position, or a "dispersed" pattern, where TSSs can be detected within a broader sequence window that is characteristic of housekeeping genes (4)"

ends with reference (4), which is completely unrelated to the content of the said sentence - indeed, I couldn't find a single paper in the list of references that deals with dispersed and focused promoters. All references should be checked carefully to make sure they are correct.

- The sentence " RNAPII turnover at PASs as part of transcriptional termination coincides with peaks of RNAPII density , perhaps indicating that RNAPII turnover near promoters may reflect transcriptional termination shortly after transcriptional initiation. " is confusing. The first part refers to PASs, the second to the sites of transcriptional initiation. The authors are trying to say that since there is known RNAPII accumulation at PASs at the ends of genes, it is possible that the accumulation of RNAPII at the pausing sites is a result of the same process, but coupled to promoter-proximal polyadenylation.

- Regarding the intragenic TSS that produce alternative gene isoforms that terminate at the gene end: please check if the TSS initiator signal at such intragenic TSS is of the same kind as that at the actual 5' ends of genes. In metazoa, while the preferred (-1,+1) dinucleotide at promoter TSSes is YR, there are intragenic TSS-like signals whose preferred starting dinucleotide is GG (see Carninci et al Nat Genet 2006).

Minor:

- "Drosophila" should be spelled in uppercase (line 93)

- line 117: Eukaryotic primary transcripts can actually be hundreds of kilobases, up to a couple of megabases (dystrophin, titin); I am aware that they are spliced long before transcriptional termination. The size of the gene and number of exons are much more relevant for providing opportunities for regulated gene isoform generation than the length of the mature transcript.

- line 217: it should be "statistically significant effect", not "affect"

Reviewer #3 (Remarks to the Author):

Thomas and colleagues have thoroughly revised their manuscript on the genomic distribution and abundance of RNA polymerase II (Pol II) initiation and termination events in *Arabidopsis thaliana*. I had three major concerns about the original manuscript. My first concern was that the size-range and gene position of short promoter-proximal RNAs (sppRNAs) needed to be confirmed via

additional northern blot experiments. The authors have successfully resolved both aspects of this concern: they detected sppRNAs with a 5'-end but not with a 3'-end MPK20 gene probe (Figure 3i), as expected, and they detected sppRNA signals at higher resolution using polyacrylamide gel electrophoresis (Figure 3j). However, in the latter figure, there is minor technical error that the authors need to fix: the Decade Marker System here consists of radiolabeled RNA molecules, so tick marks to the right of their Figure 3j membrane exposure should be labelled, "150 nt, 90 nt, 80 nt, 60 nt, 50 nt" rather than with "bp" units, which stands for base pairs and is not appropriate in this context.

My second concern was that the data presented in original Figure 1f, supporting the role of the FACT complex in repressing alternative TSSs, was confusing for the reader. The authors' revised Figure 2 is much improved. Inclusion of previously supplemental panels in revised Figure 2c and 2d is a more intuitive look at FACT complex mutant deficiencies. Moreover, revised Figure 2f now condenses the data display of intragenic TSSs detected in TIF-seq in a more approachable way. In my third concern, I had recommended that the authors moderate or remove the speculative statement, "...promoter-proximal termination is associated with plant gene expression across temperature and may contribute to temperature-dependent gene regulation." They have made adjustments to the language in their revised results and conclusions that moderate this claim. In the end, the authors data do suggest that mRNA regulation by "attenuation" could influence plant gene expression, pointing to promising avenues for future experimentation in the months and years to come.

In summary, the authors have comprehensively responded to my concerns, improved the clarity of the revised manuscript, and included additional supporting evidence along the way. I have no further reservations and recommend publication of this work, because it will influence the thinking of researchers in the fields of Pol II transcription, RNA stability and gene regulation.

The text of the comments we received by the reviewers is marked in blue, our response is marked in black.

Reviewer #2 (Remarks to the Author):

The authors have addressed most of my early concerns adequately. The new analysis of motifs is especially interesting. There are a couple of outstanding issues:

The Introduction has been expanded as other reviewers and myself suggested, but it hasn't been done in the most careful way. For example, the sentence: "The precise positions of TSSs may form a "focused" pattern with one predominant TSS position, or a "dispersed" pattern, where TSSs can be detected within a broader sequence window that is characteristic of housekeeping genes (4)" ends with reference (4), which is completely unrelated to the content of the said sentence - indeed, I couldn't find a single paper in the list of references that deals with dispersed and focused promoters. All references should be checked carefully to make sure they are correct.

Response: We apologize for the confusion. We meant to cite the excellent review by Kadonaga that reviewer #2 pointed out earlier, this review deals with focused and dispersed promoters. This reference is now included to address this point. We took the opportunity to also carefully check through all other references again. We corrected additional citations, we must have run into a problem with our citation program between several users that is now fixed in the revised manuscript.

The sentence " RNAPII turnover at PASs as part of transcriptional termination coincides with peaks of RNAPII density , perhaps indicating that RNAPII turnover near promoters may reflect transcriptional termination shortly after transcriptional initiation. " is confusing. The first part refers to PASs, the second to the sites of transcriptional initiation. The authors are trying to say that since there is known RNAPII accumulation at PASs at the ends of genes, it is possible that the accumulation of RNAPII at the pausing sites is a result of the same process, but coupled to promoter-proximal polyadenylation.

Response: Thank you for pointing this out. We have addressed this concern by following the suggestion to enhance clarity by replacing this sentence with this text to address this point: "RNAPII turnover at PASs as part of transcriptional termination coincides with peaks of RNAPII density. Since there is known RNAPII accumulation at PASs at the ends of genes, it is possible that the accumulation of RNAPII at the pausing sites is a result of the same process, but coupled to promoter-proximal polyadenylation."

Regarding the intragenic TSS that produce alternative gene isoforms that terminate at the gene end: please check if the TSS initiator signal at such intragenic TSS is of the same kind as that at the actual 5' ends of genes. In metazoa, while the preferred (-1,+1) dinucleotide at promoter TSSes is YR, there are intragenic TSS-like signals whose preferred starting dinucleotide is GG (see Carninci et al Nat Genet 2006).

Response: Thank you for this excellent suggestion. In fact, we are working on a new manuscript where we plan to include these results to give them the exposure they deserve. We hope reviewer #2 will understand that these additional analyses are beyond the scope of the current revised manuscript.

Minor:

77 - "Drosophila" should be spelled in uppercase (line 93)

Response: Done.

78 - line 117: Eukaryotic primary transcripts can actually be hundreds of kilobases, up to a couple of megabases (dystrophin, titin); I am aware that they are spliced long before transcriptional termination. The size of the gene and number of exons are much more relevant for providing opportunities for regulated gene isoform generation than the length of the mature transcript.

Response: Thank you for pointing this out. We addressed this comment by revising the text to enhance clarity: "Eukaryotic primary transcripts may extend up to a couple of megabases and provide extensive opportunities for co-transcriptional regulation of gene isoform diversity."

79 - line 217: it should be "statistically significant effect", not "affect"

Response: Done.

Reviewer #3 (Remarks to the Author):

Thomas and colleagues have thoroughly revised their manuscript on the genomic distribution and abundance of RNA polymerase II (Pol II) initiation and termination events in *Arabidopsis thaliana*. I had three major concerns about the original manuscript. My first concern was that the size-range and gene position of short promoter-proximal RNAs (sppRNAs) needed to be confirmed via additional northern blot experiments. The authors have successfully resolved both aspects of this concern: they detected sppRNAs with a 5'-end but not with a 3'-end MPK20 gene probe (Figure 3i), as expected, and they detected sppRNA signals at higher resolution using polyacrylamide gel electrophoresis (Figure 3j).

–
80 However, in the latter figure, there is a minor technical error that the authors need to fix: the Decade Marker System here consists of radiolabeled RNA molecules, so tick marks to the right of their Figure 3j membrane exposure should be labelled, "150 nt, 90 nt, 80 nt, 60 nt, 50 nt" rather than with "bp" units, which stands for base pairs and is not appropriate in this context.

Response: Thank you for finding this error. We corrected the misleading nomenclature in the figure 3j panel into nucleotides to address this concern.

My second concern was that the data presented in original Figure 1f, supporting the role of the FACT complex in repressing alternative TSSs, was confusing for the reader. The authors' revised Figure 2 is much improved. Inclusion of previously supplemental panels in revised Figure 2c and 2d is a more intuitive look at FACT complex mutant deficiencies. Moreover, revised Figure 2f now condenses the data display of intragenic TSSs detected in TIF-seq in a more approachable way.

Response: Thank you for appreciating our revisions to Figure 2.

In my third concern, I had recommended that the authors moderate or remove the speculative statement, "...promoter-proximal termination is associated with plant gene expression across temperature and may contribute to temperature-dependent gene regulation." They have made adjustments to the language in their revised results and conclusions that moderate this claim. In the end, the authors data do suggest that mRNA regulation by "attenuation" could influence plant gene expression, pointing to promising avenues for future experimentation in the months and years to come.

Response: Thank you for appreciating our new text, and for highlighting the future research implications of our findings.

In summary, the authors have comprehensively responded to my concerns, improved the clarity of the revised manuscript, and included additional supporting evidence along the way. I have no further reservations and recommend publication of this work, because it will influence the thinking of researchers in the fields of Pol II transcription, RNA stability and gene regulation.